# A primal role for the vestibular sense in the development of coordinated locomotion

**David E Ehrlich[1,2,3], David Schoppik[1,2,3]***

[1]Department of Otolaryngology, New York University School of Medicine, New York, United States; [2]Department of Neuroscience & Physiology, New York University School of Medicine, New York, United States; [3]Neuroscience Institute, New York University School of Medicine, New York, United States

**Abstract** Mature locomotion requires that animal nervous systems coordinate distinct groups of muscles. The pressures that guide the development of coordination are not well understood. To understand how and why coordination might emerge, we measured the kinematics of spontaneous vertical locomotion across early development in zebrafish (*Danio rerio*). We found that zebrafish used their pectoral fins and bodies synergistically during upwards swims. As larvae developed, they changed the way they coordinated fin and body movements, allowing them to climb with increasingly stable postures. This fin-body synergy was absent in vestibular mutants, suggesting sensed imbalance promotes coordinated movements. Similarly, synergies were systematically altered following cerebellar lesions, identifying a neural substrate regulating fin-body coordination. Together these findings link the vestibular sense to the maturation of coordinated locomotion. Developing zebrafish improve postural stability by changing fin-body coordination. We therefore propose that the development of coordinated locomotion is regulated by vestibular sensation.
DOI: https://doi.org/10.7554/eLife.45839.001

*For correspondence:
schoppik@gmail.com

**Competing interests:** The authors declare that no competing interests exist.

## Introduction

To locomote, the nervous system coordinates multiple effectors, such as the trunk and limbs or fins, that collectively generate propulsive forces and maintain body posture. For example, humans walk by using the legs to move the body forward, swinging the arms to reduce angular momentum, and using axial musculature to support the trunk (*Collins et al., 2009*). As animals mature they change the way they coordinate these effectors, a process driven both by experience and by changing motor goals (*Sporns and Edelman, 1993*; *Thelen, 1995*; *Adolph, 1997*). However, which sensations and goals guide the development of coordination is poorly understood. During development, both physical body shape and neural coordination change simultaneously (*Dickinson et al., 2000*). Understanding the constraints that guide neural control of coordination therefore requires a model in which the maturation of locomotion can be dissociated from changes in physical form (*Thelen and Ulrich, 1991*).

Development of coordination is simplified under water, where individual effectors function dissociably (*von Holst, 1973*; *Sfakiotakis et al., 1999*). Whereas forces generated while walking ultimately act through the feet, fish bodies and fins serve as independent control surfaces that need not be used in concert. For example, fish can climb in the water column using pectoral fins or body/caudal fin undulation, meaning a given climb can be executed with varying mechanics (*Aleyev, 1977*; *Webb, 2002*; *Wilga and Lauder, 2002*). These mechanics can be defined with respect to common mechanics of flight (*Figure 1A*). Bodies that move in the direction they point – like a rocket – must direct thrust upwards by pitching upwards in order to climb (*Munk, 1924*). Similarly, fish pitch

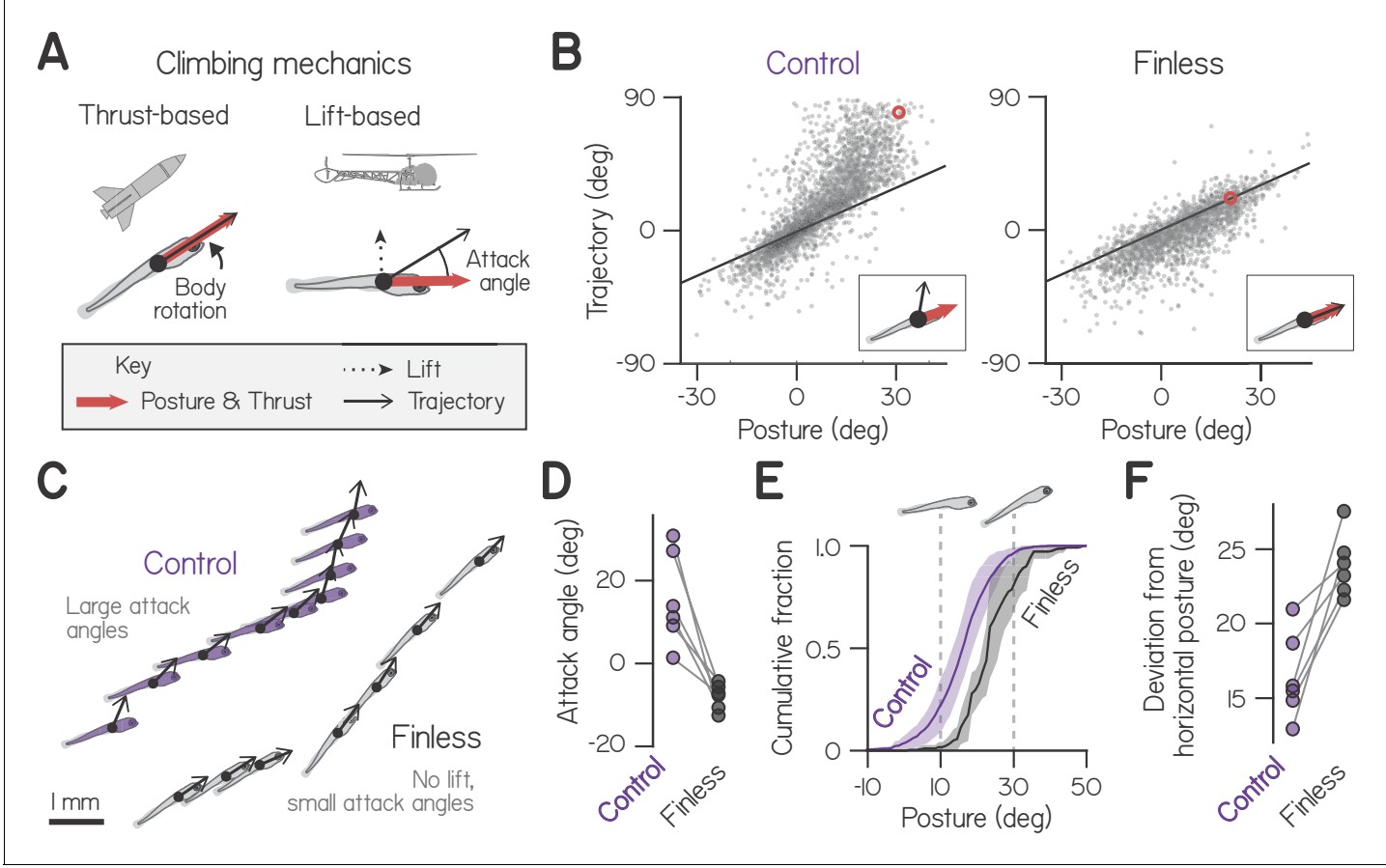

**Figure 1.** Larvae climb using bodies and pectoral fins. (A) Schematic of hydrostatic climbing mechanics. Like a rocket, a larva generates thrust in the direction it points (*top*), enabling it to generate upwards trajectories by rotating upwards to adopt nose-up postures. Complementarily, it may generate lift like a helicopter (*bottom*), creating an attack angle between trajectory and posture. (B) Trajectory of individual swim bouts as a function of posture, for control (2912 bouts) and finless larvae (1890 bouts). The unity line corresponds to 0 attack angle. Example postures and corresponding trajectories (inset) are indicated with red circles. (C) Representative epochs of climbing by one control and one finless larva, depicting posture and trajectory at the times of sequential bouts. Relative positions are to scale, but the body schematic is smaller than actual size to better highlight the trajectory. (D) Mean attack angles for control and finless siblings from six clutches (pairwise t-test, $t_5 = 4.55$, $p = 0.0061$). (E) Cumulative fractions of postures during climbs with trajectories greater than 20°, for control and finless siblings, plotted as mean ± S.D. across clutches. (F) Absolute deviation of posture from horizontal during climbs in (D) for control and finless siblings ($t_5 = 5.02$, $p = 0.0040$).

DOI: https://doi.org/10.7554/eLife.45839.002

The following figure supplements are available for figure 1:

**Figure supplement 1.** Larvae tend to sink between bouts.
DOI: https://doi.org/10.7554/eLife.45839.003

**Figure supplement 2.** Posture as a function of time during bouts by control and finless siblings.
DOI: https://doi.org/10.7554/eLife.45839.004

**Figure supplement 3.** Basic swimming statistics are unaffected by fin amputation at 1 and 3 wpf.
DOI: https://doi.org/10.7554/eLife.45839.005

upwards to direct thrust from the body/caudal fin, particularly fish with dorsoventrally symmetric bodies that generate minimal lift (*Magnuson, 1970*; *Ullén et al., 1995*). In contrast, bodies that generate lift – like a helicopter with its rotor – can remain horizontal while climbing. Fishes can produce lift using their pectoral fins (see technical note in Materials and methods) (*Aleyev, 1977*). When a fish generates lift with its pectoral fins, it moves upwards relative to its body posture, creating a non-zero attack angle (*Figure 1A*).

How fish coordinate their bodies and fins has direct consequences for balance performance. Fish can achieve a given swimming trajectory using various combinations of fin and body movements. For

a steering maneuver of a given magnitude, as the fins contribute more the body must contribute less, requiring smaller posture changes. Division of labor among the body and fins therefore impacts how posture varies, specifically in the pitch (nose-up/nose-down) axis. Many fish actively maintain horizontal posture, even during the first days of swimming (*Bagnall and McLean, 2014*; *Ehrlich and Schoppik, 2017a*), which the fins may facilitate. Pectoral fin and body movements occur synchronously in larvae (*Green et al., 2011*), but the fins appear dispensable for routine swimming in the roll and yaw axes at this stage (*Hale, 2014*). We hypothesized that fish regulate fin-body coordination in the pitch axis as they develop, learning to increasingly utilize their fins to better maintain balance as they climb.

To examine how and why fish regulate fin-body coordination across development, we studied larval zebrafish (*Danio rerio*) as they spontaneously climbed in the water column. We compared groups of siblings, or clutches, throughout the larval stage (3–30 days post-fertilization, dpf) (*Parichy et al., 2009*). Larvae locomote in discrete bouts approximately once per second, simplifying kinematic analysis (*Ehrlich and Schoppik, 2017a*; *Marques et al., 2018*). We found that larvae climbed with steeper trajectories than would be predicted from posture alone – evidence they were actively generating lift. After fin amputation, larvae no longer generated lift. We found that larvae at all ages exhibited correlated fin-driven lift and body rotations, strong evidence for active fin-body coordination. Consistently, we found that fin-body coordination was abolished in vestibular mutants with an impaired sense of balance (*Roberts et al., 2017*) and perturbed by cerebellar lesions.

Developing larvae regulated fin-body coordination to rely increasingly on their fins during climbs. Consequentially, older larvae were observed to climb with balanced postures closer to horizontal. To understand why larvae at different ages coordinated their fins and bodies in different ways, we built a model to explore the trade-offs incurred by a drive to balance. Simulations showed that more mature coordination, dominated by pectoral fins, improved balance but cost greater effort (Appendix). We conclude that developmental changes to coordination help stabilize posture when climbing. Together, these data suggest that the vestibular sense guides the development of coordinated locomotion.

## Results

### Larvae use pectoral fins to balance while climbing

First, we examined climbing kinematics of zebrafish late in the larval stage, from 2912 swim bouts captured from 45 larvae across 6 clutches at 3 weeks post-fertilization (wpf). Larvae tended to pitch upwards in order to swim upwards, yielding a positive correlation of trajectory and pitch-axis posture that reflected thrust-based climbing (*Figure 1B*; Spearman's $\rho = 0.81$). In addition, larvae often swam with positive attack angles (defined here as the difference between trajectory and posture), swimming more upwards than they oriented, reflecting the production of lift. Larvae exhibited positive attack angles preferentially when climbing, in 92.5% of bouts with upwards trajectories (1866/2016). By comparison, larvae exhibited positive attack angles in only 13.8% of bouts when diving (124/896). Larvae therefore generate lift specifically when pitched upwards to climb.

We hypothesized that larvae generated lift using their pectoral fins, because they tend to abduct the fins while swimming (*Thorsen et al., 2004*; *Green and Hale, 2012*) and did so when propelling upwards (*Videos 1* and *2*). When we amputated the pectoral fins and recorded 1890 bouts after 4–5 hr recovery, we found that positive attack angles were largely abolished (*Figure 1B and C*). Control larvae exhibited attack angles of 15.6° on average, compared to −8.0° for finless siblings (*Figure 1D*; n = 6 groups of 6–8 finless larvae, pairwise t-test: $t_5$ = 4.55, p = 0.0061). A finless larva simply propelled in the direction it pointed, exhibiting a trajectory that closely approximated its posture,

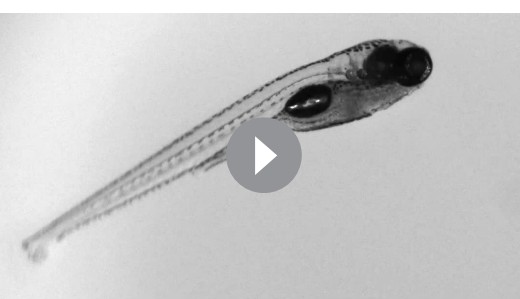

**Video 1.** Lateral view of a freely-swimming, two wpf larva producing 4 bouts of upwards motion interleaved by periods of slow sinking.
DOI: https://doi.org/10.7554/eLife.45839.006

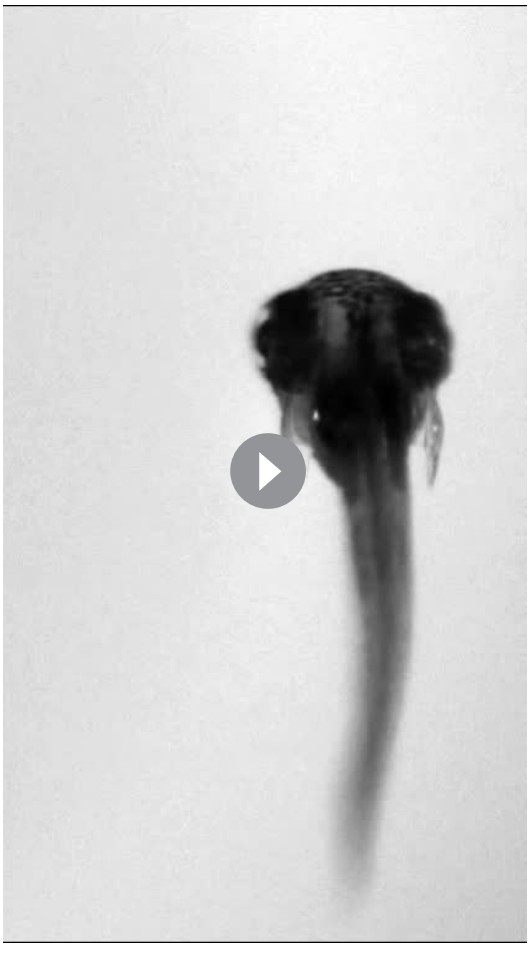

**Video 2.** View down the long axis of a freely-swimming, two wpf larva producing 5 bouts of upwards motion with visible pectoral fin abduction.
DOI: https://doi.org/10.7554/eLife.45839.007

albeit with a minor downward bias. Accordingly, finless larvae never made the near vertical climbs of control siblings (*Figure 1B*). Small negative attack angles exhibited by finless larvae are consistent with a slight negative buoyancy (*Stewart and McHenry, 2010*) and an observed tendency to sink between propulsive bouts (*Figure 1—figure supplement 1*; *Video 1*). We conclude larvae at 3 weeks post-fertilization (wpf) used their pectoral fins to generate lift.

Importantly, larvae effected thrust- and lift-based climbing independently. Pectoral fin amputation did not influence body rotations in the pitch axis (*Figure 1B*, *Figure 1—figure supplement 2*) or swimming more generally (*Figure 1—figure supplement 3*; consistent with *Green et al., 2011*). Specifically, amputation had no significant effect on bout maximum speed (paired t-test, $p > 0.05$, $t_{10} = -1.14$), displacement ($t_{10} = -1.66$), rate ($t_{10} = 0.23$), or absolute pitch-axis rotation ($t_{10} = 1.25$). We conclude that larvae do not require their pectoral fins to pitch upwards and climb, presumably instead rotating using the body and caudal fin (*Ullén et al., 1995*; *Bagnall and Schoppik, 2018*).

We hypothesized that larval pectoral fins are well suited for generating lift without torque because they attach near the body center of mass (*Drucker and Lauder, 2002*). Pectoral fins would therefore act over a small moment arm to generate torques in the pitch axis, making those torques small. We measured the rostrocaudal positions of pectoral fin attachment from 15 larvae and compared those to morphometrically estimated positions of the center of mass (*Ehrlich and Schoppik, 2017a*). Indeed, the pectoral fins attached consistently near the center of mass, on average $0.056 \pm 0.007$ body lengths rostrally (*Figure 2—figure supplement 1*). The position of the pectoral fins in larval zebrafish may therefore facilitate dissociation of lift- and thrust-based climbing, enabling larvae to specifically use pectoral fins to produce lift without causing pitch-axis body rotation.

Following pectoral fin amputation, larvae compensated for loss of lift by changing their posture. Specifically, larvae rotated further from horizontal in order to climb (*Figure 1C*). In order to produce climbs steeper than 20°, finless larvae pitched significantly further upwards than control siblings; they adopted postures of 23.5° compared to 16.5° (*Figure 1E and F*; pairwise t-test, $t_5 = 5.02$, $p = 0.0040$). Consistently, finless larvae were unable to produce steep climbs at horizontal postures, unlike control siblings. We conclude that use of the pectoral fins for climbing facilitates balance, enabling larvae to maintain postures near horizontal.

## Larvae coordinate fins and bodies to climb

Larvae could facilitate climbing by combining independent lift- and thrust-based mechanisms (*Figure 2A*). Pairing fin-mediated lift with upwards posture changes would yield synergistic climbing effects. Conversely, lift from fins would interfere with diving produced by downwards posture changes. If larvae concertedly use both their fins and bodies to climb and dive, we would expect attack angles and postural control to be correlated.

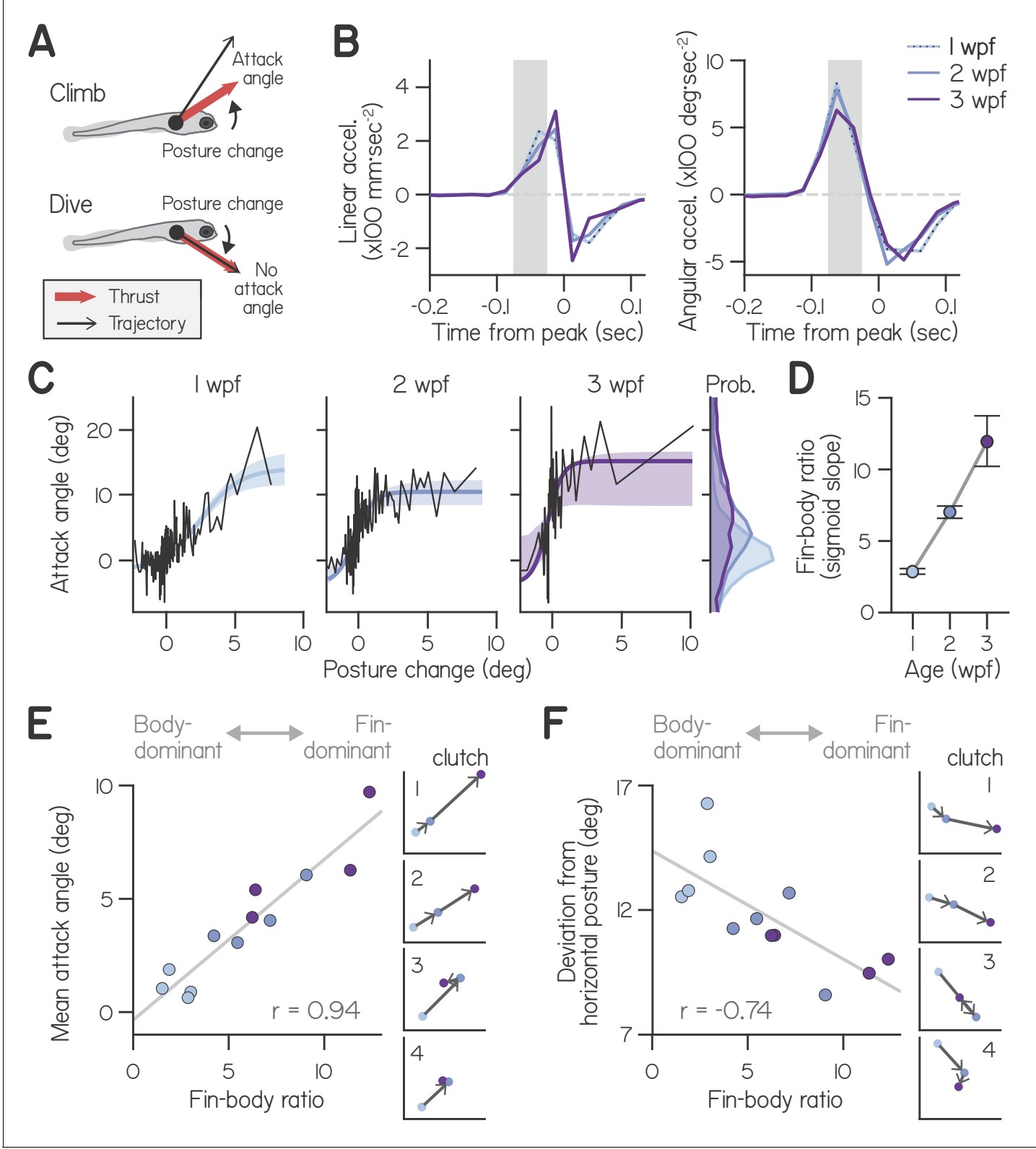

**Figure 2.** Development of fin-body coordination. (**A**) Schematic of fin-body coordination for climbing. Positive posture changes are paired with positive attack angles and negative body rotations with no attack angle, reflecting exclusion of the fins. (**B**) Mean linear and angular acceleration during swim bouts at 1, 2, and 3 weeks post-fertilization (wpf), temporally aligned to peak linear speed (time 0). The window used to compute posture change is highlighted in gray. (**C**) Attack angle as a function of posture change for bouts at 1, 2, and 3 wpf, with cropped attack angle probability distributions

*Figure 2 continued on next page*

*Figure 2 continued*

(*right*). Data plotted as means of equally sized bins (black lines) and superimposed with best-fit sigmoids and their bootstrapped S.D. (D) Fin-body ratio, defined as the maximal slope of best-fit sigmoid to attack angle and posture change, is plotted with 95% confidence intervals as a function of age. (E,F) Mean attack angle (E) and absolute deviation from horizontal (F) for each clutch and age, evaluated over 48 hr, are plotted as functions of fin-body ratio with Pearson's correlation coefficients (*r*; p=5.6E-6 for attack angle and p=6.3E-3 for deviation from horizontal). Small values convey body-dominant synergies, while large values convey fin-dominant synergies. Developmental trajectories for the four individual clutches are plotted on identical axes (*right*).

DOI: https://doi.org/10.7554/eLife.45839.008

The following figure supplements are available for figure 2:

**Figure supplement 1.** Pectoral fins and bodies grow proportionally.

DOI: https://doi.org/10.7554/eLife.45839.009

**Figure supplement 2.** Clutch- and age-specific fin bias.

DOI: https://doi.org/10.7554/eLife.45839.010

To understand how developing larvae coordinated their fins and bodies, we examined concurrent control of these effectors during bouts. We measured swimming at 1, 2, and 3 wpf across four clutches (3552, 2326, and 693 bouts, respectively). Additionally, we examined their newly-swimming siblings at 4 days post-fertilization and found that fin use was indistinguishable from that at 1 wpf (4004 bouts, *Table 1*).

First, we assessed how larvae used their bodies to direct thrust. Because larvae actively control their posture during swim bouts, we reasoned that they may acutely change posture in pitch to direct thrust up or down (*Ehrlich and Schoppik, 2017a*; *Ehrlich and Schoppik, 2017b*). To assess whether larvae changed posture before generating thrust, we compared the timing of angular and linear accelerations during spontaneous swim bouts. We found that larvae at all ages produced large, pitch-axis angular acceleration preceding and during thrust generation, when they accelerated forwards (*Figure 2B*). Angular acceleration lasted approximately 100 ms and peaked 62.5 ms before larvae ceased generating thrust and began linear deceleration. We defined the steering-related posture change of a bout from 25 to 75 ms before linear deceleration, and observed that all larvae exhibited comparable posture changes (*Table 1*; Two-way ANOVA, main effect of age: $F_{2,6} = 2.21$, p = 0.19; main effect of clutch: $F_{3,6} = 1.89$, p = 0.23).

Larvae used their fins and bodies synergistically, particularly during steep climbs. Larvae at all ages exhibited positively correlated attack angles and posture changes (*Figure 2C*), with Spearman's correlation coefficients of 0.27 ± 0.08 at 1 wpf (mean ± S.D. across clutches), 0.38 ± 0.13 at 2 wpf, and 0.37± 0.14 at 3 wpf (*Table 1*). In particular, larvae paired large, upwards posture changes (>5˚) with positive attack angles; of 210 bouts with such posture changes across all ages, 193 (92%) had positive attack angles (binomial test: p = 1.5E-21, given 63.4% of all bouts had positive attack angles).

To confirm that young larvae, like older larvae, generated positive attack angles using pectoral fins, we examined the effects of fin amputation at 1 wpf. Large attack angles (greater than 15˚) were rare but observable in control larvae at 1 wpf (3.8%, 81/2652 bouts). In contrast, large attack angles were nearly abolished in siblings following pectoral fin amputation (0.6%, 16/2630 bouts; n = 6 groups of 7–8 finless larvae, pairwise t-test: $t_5 = 4.40$, p = 0.0070). We conclude larvae at all ages coordinated their fins and bodies in order to climb.

## Developing larvae regulate fin-body coordination

Correlations between body and fin actions changed with age. Specifically, younger larvae paired a given posture change with smaller attack angles (*Figure 2C*). As a first pass, we quantified the ratio of attack angles to posture changes during shallow climbs (with posture changes from 0˚ to 3˚) using a robust slope estimate; with age, the ratio of attack angles to posture changes nearly tripled, from 0.71:1 at 1 wpf to 1.61:1 at 2 wpf and 2.00:1 at 3 wpf. We conclude that older larvae produced small climbs with greater contribution from the pectoral fins.

Larvae at all ages made the steepest climbs similarly, pairing the largest posture changes (5˚−10˚) with comparable attack angles (10˚−20˚, on average). Attack angles reached an asymptote as a function of posture change (*Figure 2C*), which we interpret as a physical constraint on attack angle; after maximizing attack angle, larvae could only climb more steeply by rotating further upwards. Because

**Table 1.** Empirical and simulated swimming properties and morphological measurements as a function of age.
Sigmoid parameters refer to the best-fit logistic function to attack angle vs. body rotation (*Equation 1*), comprising 4 degrees of freedom. $\rho$: Spearman's correlation coefficient.

| Variable | Unit | 4dpf | 1wpf | 2wpf | 3wpf |
|---|---|---|---|---|---|
| Mean attack angle | deg | 0.87 | 1.02 | 4.78 | 8.11 |
| $R^2$ of trajectory and posture | - | 0.86 | 0.91 | 0.78 | 0.66 |
| Deviation from horizontal | deg | 13.91 | 14.08 | 11.91 | 11.30 |
| Swim bout peak speed | mm/s | 11.2 | 13.6 | 13.4 | 14.1 |
| Swim bout duration | s | 0.093 | 0.082 | 0.087 | 0.106 |
| Swim bout displacement | mm | 1.24 | 1.29 | 1.24 | 1.43 |
| Mean bout posture change | deg | 0.10 | −0.23 | 0.24 | 0.21 |
| Standard deviation of bout posture change | deg | 2.21 | 1.84 | 1.84 | 2.10 |
| $\rho$ of attack angle and body rotation | - | 0.305 | 0.269 | 0.379 | 0.368 |
| Proportion of climbs with trajectory > 20° | - | 0.26 | 0.30 | 0.34 | 0.43 |
| Body length | mm | 4.18 | 4.26 | 5.57 | 7.92 |
| Pectoral fin length | mm | 0.41 | 0.42 | 0.61 | 0.90 |
| Fin distance anterior to COM | mm | 0.27 | 0.22 | 0.27 | 0.44 |
| Sigmoid amplitude $\gamma_{max}$ | deg | 19.28 | 15.71 | 14.30 | 18.79 |
| Sigmoid vertical location, $\gamma_0$ | deg | −3.00 | −1.59 | −3.72 | −3.56 |
| Sigmoid horizontal location, $r_{rest}$ | deg | −0.77 | −0.42 | −1.75 | −1.51 |
| Sigmoid slope, $k \cdot \gamma_{max}/4$ | - | 2.76 | 2.89 | 7.03 | 12.01 |
| Goodness-of-fit ($R^2$) for 4-parameter sigmoid ($k, \gamma_{max}, \gamma_0, r_{rest}$) | - | 0.195 | 0.115 | 0.113 | 0.087 |
| Goodness-of-fit ($R^2$) for 1-parameter sigmoid ($k$) | - | 0.193 | 0.109 | 0.092 | 0.086 |
| Empirical fin bias, $\hat{\alpha}$ | - | 0.73 | 0.74 | 0.88 | 0.92 |
| Balance weight in cost function, β | - | 0.12 | 0.12 | 0.18 | 0.32 |

DOI: https://doi.org/10.7554/eLife.45839.011

larvae at all ages exhibit comparable swimming speeds (*Table 1*, Two-way ANOVA with main effects of age and clutch, $F_{2,6} = 0.94$, p = 0.44), similar maximal attack angles reflect similar dorsal acceleration from the pectoral fins. The same fin-mediated acceleration would require greater force generation, given increasing body mass. Consistently, pectoral fins grew with age but maintained similar proportional lengths to the body at 1 wpf (0.098 ± 0.010 body lengths), 2 wpf (0.110 ± 0.008), and 3 wpf (0.114 ± 0.008, n = 15; *Figure 2—figure supplement 1*, *Table 1*; One-way ANOVA, $F_{2,42} = 13.19$, p = 3.60E−5). These data suggest that larvae do not become physically more capable of climbing with the fins as they develop.

Instead, developing larvae changed how they distributed labor among the body and fins. Older larvae used the largest attack angles to climb on a greater proportion of bouts than younger larvae. Larvae at 3 wpf paired 2–3° posture changes with large 15.2° attack angles; although larvae at 1 and 2 wpf were capable of generating large attack angles, they paired 2–3° posture changes with attack angles of 5.8° and 8.8°, respectively. Furthermore, older larvae exhibited near-maximal fin use (>10° attack angle) on a far greater proportion of bouts (5.4% at 1 wpf, 19.9% at 2 wpf, and 38.7% at 3 wpf). On average, larvae exhibited gradually increasing attack angles with age (*Figure 2C*, *marginals*; main effect of age by two-way ANOVA, $F_{2,6} = 9.46$, p = 0.014), with significantly smaller angles at 1 wpf (1.02°) than 3 wpf (8.11°, p = 0.004; Tukey's posthoc test). Together, these data suggest that changes in fin-body coordination, rather than physical ability, account for the nearly 8-fold increase in average attack angles from 1 to 3 wpf.

To model how attack angle varied as a function of posture change, we fit data with sigmoids (*Figure 2C*). We used logistic functions comprising four parameters: one to capture sigmoid amplitude, another for sigmoid steepness, and two for location (see Materials and methods). Three parameters (for sigmoid amplitude and location) did not significantly differ across ages, further

support for the hypothesis that fin capability is constrained across early development (*Table 1*). In contrast, the dimensionless steepness parameter significantly varied with age.

Sigmoid steepness captured fin-body coordination throughout development, reflecting increasing engagement of the fins relative to the body. We termed the maximal slope of the sigmoid the 'fin-body ratio.' This ratio increased more than four-fold with age (from 2.9 at 1 wpf to 7.0 at 2 wpf and 12.0 at 3 wpf) after fixing the remaining three parameters at their means across ages (*Figure 2D*). We conclude that larvae at all ages were capable of the same range of attack angles, but older larvae favored the fins when climbing, pairing large attack angles with smaller posture changes.

The fin-body ratio was sufficient to describe variations in climbing behavior across clutches. We measured fin-body ratio for individual clutches at each age, combining data over two successive recording days for good sigmoid fits (*Figure 2—figure supplement 2*). The fin-body ratio exhibited a significant positive correlation with mean attack angle (*Figure 2E*, Pearson's $r = 0.94$, $p = 5.6E-6$) but not mean trajectory ($r = 0.29$, $p = 0.28$) or the frequency of steep climbs ($r = 0.40$, $p = 0.13$). Furthermore, the fin-body ratio reflected clutch differences in development of fin use; only clutches with increased fin-body ratios from 2 to 3 wpf displayed increased attack angles (*Figure 2E*). These data suggest larvae swam with more lift while making the same climbs simply by biasing the composition of fin-body coordination toward the fins.

Beyond simply describing the relationship between fin and body actions, the fin-body ratio was related to balance performance. Larvae without fins exhibited worse balance in the pitch axis by adopting postures further from the horizontal (*Figure 1E and F*). In intact fish, pitch-axis posture was correlated with the fin-body ratio. Specifically, this ratio exhibited a significant negative correlation with the absolute deviation from horizontal posture (*Figure 2F*, $r = -0.74$, $p = 6.3E-3$). During climbs steeper than 20°, larvae at 1 wpf adopted postures pitched significantly more upwards (28.0°; two-way ANOVA, main effect of age: $F_{2,6} = 25.29$, $p = 0.0012$) than larvae at 2 wpf (19.7°; Tukey's test, $p = 0.0040$) or 3 wpf (17.7°; Tukey's test, $p = 0.0013$). The fin-body ratio also reflected clutch differences in development of balance; the lone clutch exhibiting a large decrease in fin-body ratio from 2 to 3 wpf (from 9.0 to 6.4) also exhibited worse balance, with larger deviation from horizontal at 3 wpf (11.0°) than at 2 wpf (8.6°; *Figure 2F*, clutch 3). Regardless of age, larvae that preferentially used their fins to climb remained nearer horizontal. We conclude that a single parameter, the fin-body ratio, captures variability of fin-body coordination across development and its consequences for balance.

## Fin-body coordination requires vestibular sensation

To confirm that correlated fin and body actions arose due to coordination rather than biomechanics, we tested whether fin-body correlations were influenced by sensory perturbation. Specifically, we hypothesized that vestibular sensation promoted fin-body coordination, because coordination was correlated with measures of balance performance (*Figures 1E* and *2F*) and vestibular stimuli can elicit pectoral fin movements (*Timerick et al., 1990*). We examined swimming in one wpf larvae with genetic loss of function of utricular otoliths, sensors of head/body orientation relative to gravity (*Braemer and Braemer, 1958*). Utricular otolith formation is delayed from 1 to 14 dpf by loss-of-function mutation of *otogelin* (*Mo et al., 2010*; *Roberts et al., 2017*) (*Figure 3A*). *otogelin* is expressed exclusively in cells in the otic capsule (*Cohen-Salmon et al., 1997*) where it is required for tethering of the otolith to the macula (*Stooke-Vaughan et al., 2015*).

We found that correlated fin- and body-mediated climbing was abolished in *otogelin - /-* larvae. Mutants exhibited no correlation of attack angle and posture change across 3656 bouts (*Figure 3B*; Spearman's $\rho = 0.03$, $p = 0.051$; $n = 56$ larvae from five clutches). In contrast, control siblings with functioning utricles exhibited a significant, positive correlation of attack angle and posture change across 4767 bouts ($\rho = 0.15$, $p = 2.01E-26$). The correlation between attack angle and posture change was significantly lower in *otogelin - /-* larvae than controls from the same clutch, assessed by pairwise t-test ($t_4 = 4.01$, $p = 0.016$). Accordingly, the fin-body ratio was significantly smaller for mutants than controls and indistinguishable from zero (*Figure 3C*; with 95% CI: $0.07 \pm 0.09$ vs. $1.11 \pm 0.13$). Furthermore, *otogelin -/-* larvae failed to pair large, upwards posture changes (>5°) with positive attack angles; of 144 bouts with such posture changes, only 90 had positive attack angles (binomial test: $p = 0.215$, given that 66.0% of bouts had positive attack angles). By comparison, control siblings exhibited positive attack angles on 112 of 141 bouts with large, upwards posture changes ($p = 5.2E-8$, given that 57.9% of bouts had positive attack angles). We conclude that correlated

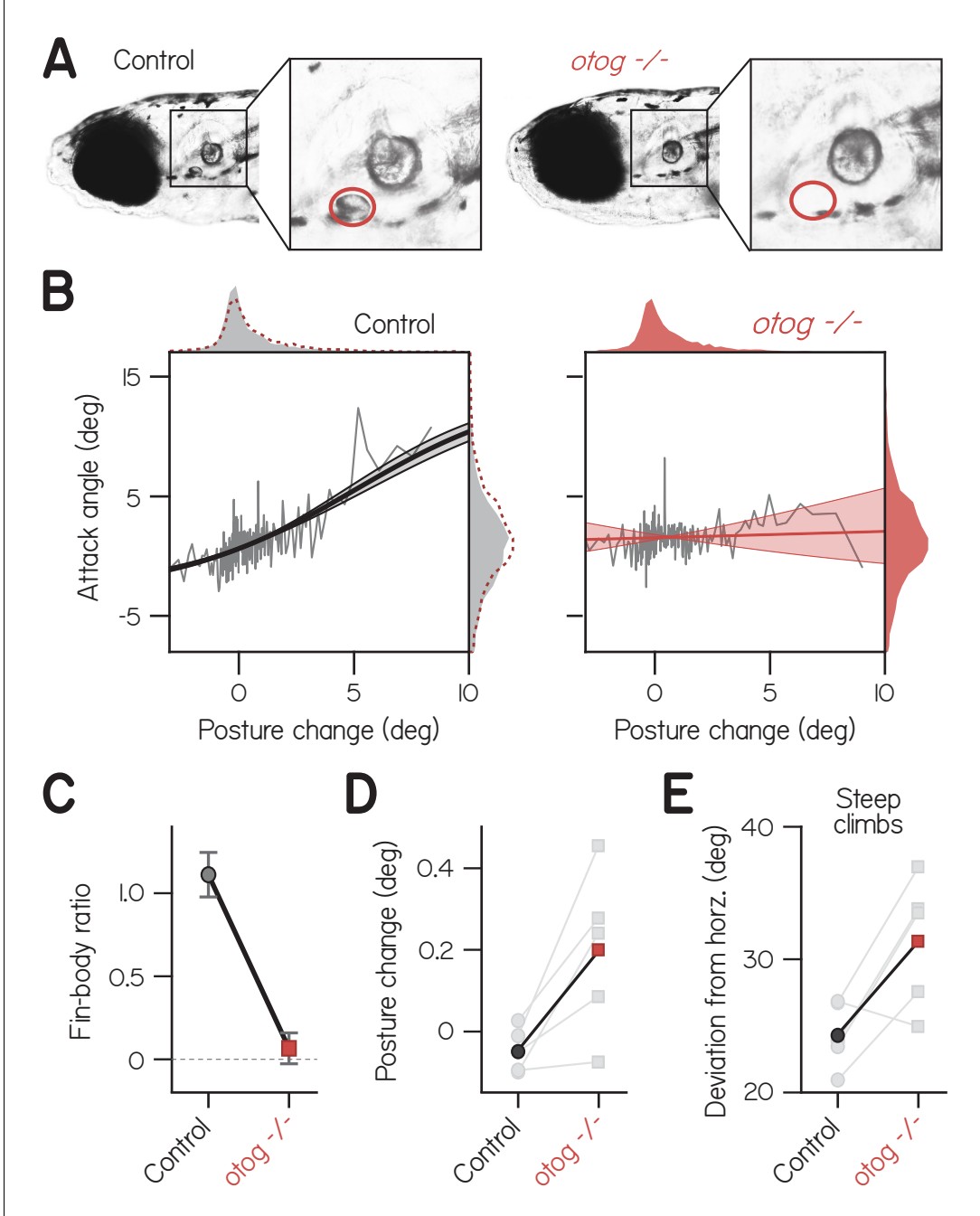

**Figure 3.** Fin-body coordination is abolished by peripheral vestibular lesion. (**A**) Representative lateral photomicrographs, one of a larva with typical development of utricular (anterior) otoliths (*top*, control: wild-type or heterozygous for *otogelin*) and another of its sibling lacking utricular otoliths (*bottom*, *otogelin -/-*). Utricle position is encircled in red. (**B**) Attack angle as a function of posture change for bouts by control larvae (4767 bouts) and *otogelin -/-* siblings (3656 bouts). Data plotted as means of equally sized bins (gray lines) superimposed with best-fit sigmoids and their bootstrapped S.D. Marginals show cropped probability distributions, with *otogelin -/-* marginals superimposed on control data as dashed lines. (**C**) Fin-body ratio, or the maximal slope of best-fit sigmoid, plotted with 95% confidence interval. (**D**) Median posture change during bouts by individual clutches (gray) and their means. Pairwise t-test, $t_4$ = 3.13, p = 0.035. (**E**) Mean deviation of posture from horizontal during steep climbs (> 20°). Pairwise t-test, $t_4$ = 3.02, p = 0.039.

DOI: https://doi.org/10.7554/eLife.45839.012

actions of the fins and body are generated by the nervous system using sensory information, and therefore constitute coordination.

As expected from the restricted pattern of gene expression, deficits in *otogelin -/-* larvae appeared to be specific to the sensory periphery. Mutants have no reported defects in the central nervous system (*Roberts et al., 2017*) and appeared morphologically unaffected. We observed typical morphology of the body and pectoral fins (0.43 ± 0.03 mm fin length vs. 0.42 ± 0.04 mm for controls; n = 15; $t_{28}$ = 0.80, p=0.43; *Table 2*). Consistently, distributions of attack angles were comparable for *otogelin -/-* larvae (1.6 ± 5.2°) and siblings (1.1 ± 6.6°; *Figure 3B*, marginals), suggesting they are capable of generating lift with the fins but fail to do so when climbing with the body. Validating direct comparison of climbing kinematics between mutants and control siblings, we found that *otogelin -/-* larvae made steep climbs (>20°) as frequently as control siblings with utricles (35 ± 13% vs. 23 ± 9% for controls; pairwise t-test, $t_4$ = 1.89, p = 0.13) and could generally balance, orienting approximately horizontally on average in the light (8.35°). Gross swimming properties were also similar between *otogelin -/-* larvae and controls (*Table 3*).

Like finless larvae, vestibular mutants that failed to coordinate their fins and bodies deviated further from horizontal. Posture changes by *otogelin -/-* larvae were directed significantly more upwards than those by control siblings (*Figure 3D*; pairwise t-test, $t_4$ = 3.13, p = 0.035), which presumably compensates for less lift while climbing. Accordingly, *otogelin -/-* larvae exhibited significantly larger deviations from horizontal during climbs steeper than 20° (*Figure 3E*; 31.4 ± 4.9° vs. 24.3 ± 2.5° for controls; $t_4$ = 3.02, p = 0.039). We conclude that loss of fin-body coordination necessitates larger deviations from horizontal to climb.

## The cerebellum facilitates fin-body coordination

The cerebellum is canonically involved in motor coordination and vestibular learning (*Thach et al., 1992*; *du Lac et al., 1995*) and cerebellar circuitry is largely conserved among vertebrates (*Altman and Bayer, 1997*; *Hashimoto and Hibi, 2012*). We hypothesized that fin-body coordination is regulated by the cerebellum and specifically by Purkinje cells, efferent neurons of the cerebellar cortex that directly innervate the vestibular nuclei (*Hashimoto and Hibi, 2012*; *Hamling et al., 2015*). To test this hypothesis, we lesioned Purkinje cells using the photosensitizer, KillerRed (*Del Bene et al., 2010*), targeted using the gal4:UAS system with a selective driver in Purkinje cells Tg(aldoca:GAL4FF) (*Takeuchi et al., 2015*). After light exposure, we measured swim bout kinematics at 1 wpf (602 from six larvae) and compared them to bouts from unexposed KillerRed+ siblings (408 from 10 larvae). Swim kinematics were largely unaffected by Purkinje cell lesions (*Table 3*) but postures tended nose-up (17.7 ± 20.6° vs. 8.3 ± 17.5° for controls).

Fin-body coordination was perturbed in larvae with Purkinje cell lesions. These larvae exhibited more positive attack angles than controls (*Figure 4A*; 3.83° vs. 0.82°; Kolmogorov-Smirnov test, p = 1.6E-11), with comparable values to wild-type larvae a week older. Specifically, larvae with lesions exhibited positive attack angles during bouts with nose-down posture changes. Typically, larvae at all ages suppressed positive attack angles while rotating nose down. Given that positive attack angles reflect lift generation by the fins, and nose-down posture changes direct thrust downwards, such fin and body actions are conflicting.

To determine the probability that larvae performed conflicting fin-body actions, we identified bouts with nose down posture changes (< −1°) and measured the proportion with attack angles more positive than baseline (−1.59°, from wild-type fits at 1 wpf, *Table 1*). Control larvae performed conflicting actions significantly less frequently than chance (*Figure 4B*; 0.429 ± 0.071, with 95% CI),

---

**Table 2.** Morphology of *otog-/-* larvae and control siblings (*otog+/-* and *otog+/+* with utricles). Data listed as mean ± S.D.

| Variable | Unit | Otog-/- | Control |
|---|---|---|---|
| Body length | mm | 4.52 ± 0.32 | 4.53 ± 0.23 |
| Pectoral fin length | mm | 0.43 ± 0.03 | 0.42 ± 0.04 |
| Pectoral fin length | % body length | 9.6 ± 0.6 | 9.3 ± 0.8 |

DOI: https://doi.org/10.7554/eLife.45839.013

**Table 3.** Swim bout properties for *otog-/-* and *Tg(aldoca:GFF);Tg(UAS:KillerRed)* larvae.
Data listed as mean ± S.D.

| Variable | Unit | *Otog-/-*, no utricle | Utricle control | *Aldoca::KR* lesioned | *Aldoca::KR* control |
|---|---|---|---|---|---|
| Maximum linear speed | mm·s$^{-1}$ | 12.4 ± 4.5 | 12.0 ± 4.3 | 10.7 ± 4.3 | 12.7 ± 4.5 |
| Duration | s | 0.084 ± 0.034 | 0.079 ± 0.033 | 0.109 ± 0.070 | 0.120 ± 0.051 |
| Displacement | mm | 1.23 ± 0.61 | 1.13 ± 0.54 | 1.25 ± 0.75 | 1.62 ± 0.71 |
| Maximal pitch-axis angular speed | deg·s$^{-1}$ | 98.7 ± 73.0 | 90.1 ± 69.0 | 84.4 ± 54.0 | 100.6 ± 61.4 |
| Inter-bout interval | s | 1.22 ± 1.37 | 1.09 ± 1.03 | 2.09 ± 2.30 | 2.12 ± 2.80 |

DOI: https://doi.org/10.7554/eLife.45839.014

and larvae with lesions performed conflicting actions significantly more frequently than chance (0.642 ± 0.121). Larvae with lesions were also significantly more likely to perform synergistic fin-body actions, pairing positive attack angles with nose-up posture changes (*Figure 4C*; 0.844 ± 0.053 of swim bouts vs. 0.736 ± 0.050 for controls).

Importantly, larvae with lesions exhibited more positive attack angles when making larger magnitude posture changes, be they nose-up or nose-down (*Figure 4A*). In order to quantify the relative magnitude of attack angles to both positive and negative posture changes, we modeled these data as the sum of two sigmoids, one of which was reflected about the vertical axis. Best-fit sigmoids captured the tendency to engage the fins during large positive and negative posture changes. For larvae with Purkinje cell lesions, the largest magnitude slope of the nose-down sigmoid significantly differed from 0 (*Figure 4D*; −4.09). Furthermore, that slope did not significantly differ in magnitude from the slope of the best-fit nose-up sigmoid (*Figure 4E*; 6.50). In contrast, the double sigmoid was overparameterized for fitting control data, and the maximal slope of the nose-down sigmoid did not differ from 0 (*Figure 4D*; −0.22; see Materials and methods). Finally, the slope of the nose-up sigmoid was significantly larger for larvae with Purkinje cell lesions compared to controls (*Figure 4E*; 6.50 vs. 1.69). We conclude that the cerebellum actively suppresses fin-mediated lift generation during pitch-axis steering. Our data suggest a dual role for cerebellar regulation of fin-body coordination: to bias division of labor towards the body, and to prevent the production of conflicting fin-body actions (generating lift during body-mediated diving).

## Discussion

Here, we used a new model to study coordinated movements and discovered a fundamental role for the vestibular sense in the development of locomotor coordination. First, we demonstrated that to climb, zebrafish larvae used upward-orienting body rotations and lift-producing pectoral fin actions. Larvae actively coordinated two independent effectors, the trunk and the pectoral fins, to locomote upward. As they developed, larvae came to match larger fin actions with smaller body rotations. Younger larvae were capable of the same independent fin and body actions as older larvae, suggesting fin-body coordination matures due to neural rather than physical changes. Because manipulation and natural variation of coordination impacted posture, we hypothesized that coordination was regulated by sensations about posture transduced by the vestibular system. Mutants with deficient vestibular sensation did not coordinate the trunk and fins despite performing similar body and fin actions, linking the vestibular sense to coordination. Cerebellar lesions uncovered conflicting fin-body actions, revealing a neural substrate for regulation of fin-body coordination. Taken together, our data show how the vestibular sense comes to shape the development of coordinated locomotion.

Our data establish a locomotor function for pectoral fins in larval zebrafish. Previous work using larvae examined pectoral fin kinematics during yaw and roll turns (*Green et al., 2011*; *Hale, 2014*). Few differences were observed in yaw and roll between wild-type fish and mutants lacking pectoral fins. Instead, pectoral fin movements *between* bouts led the authors to propose the intriguing hypothesis that pectoral fin movements played a role in respiration. Complementarily, we find that larvae use their pectoral fins during climbing bouts to generate lift. These data establish a novel locomotor function for fins in larval zebrafish, providing a more complete picture of their utility.

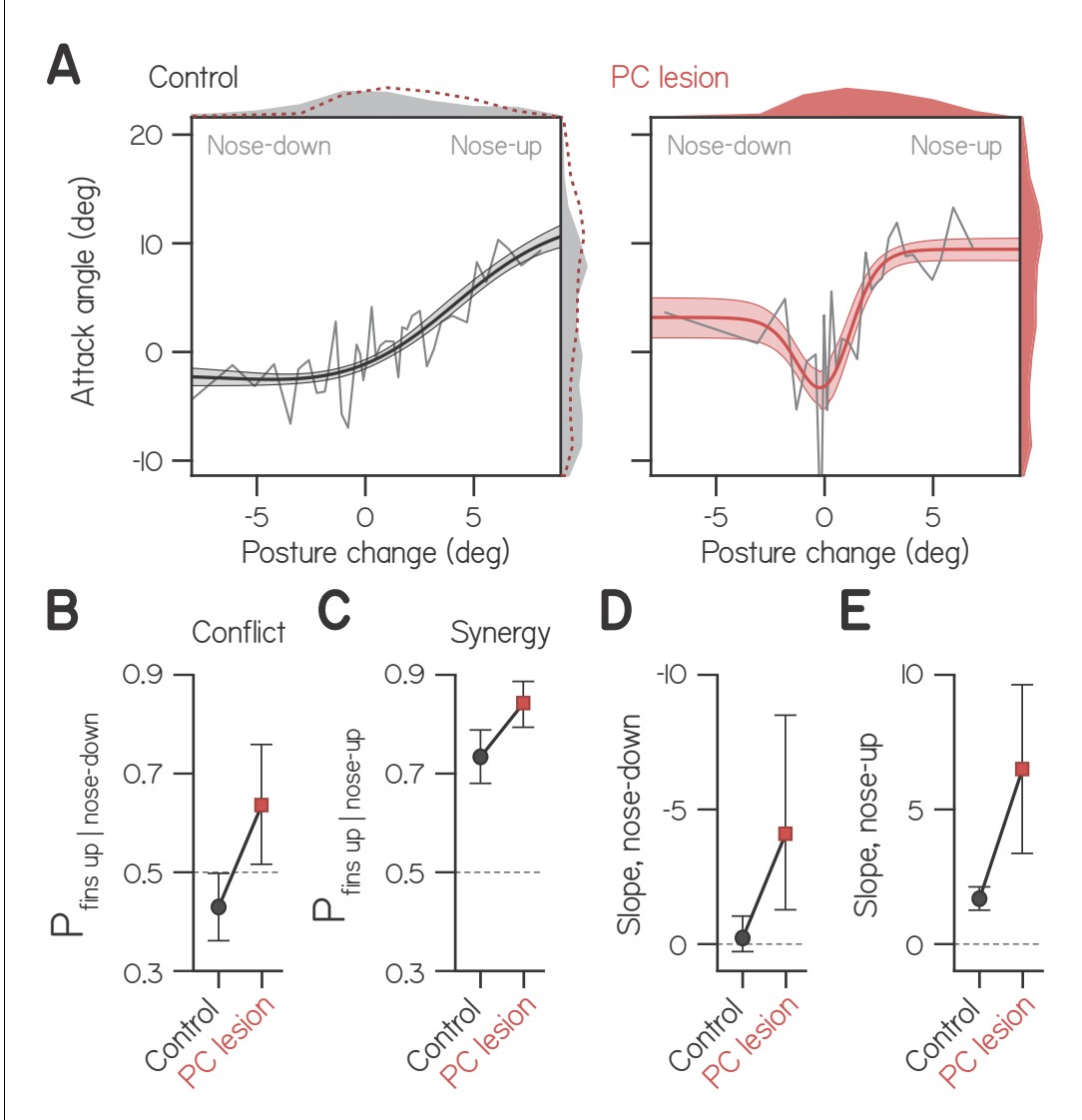

**Figure 4.** Cerebellar lesion impairs fin-body coordination. (A) Attack angle as a function of posture change for bouts by control larvae (602 bouts) and siblings with lesioned Purkinje cells (408 bouts). Data plotted as means of equally-sized bins (gray lines) superimposed with best-fit sum of two sigmoids and their bootstrapped S.D. Marginals show cropped probability distributions, with marginals from lesioned larvae superimposed on control data as dashed lines. (B) Proportion of bouts with attack angles more positive than 1 wpf baseline ($-1.59°$) given nose-down posture change ($< -1°$), with bootstrapped 95% CI. (C) Proportion of bouts with attack angles more positive than 1 wpf baseline ($-1.59°$) given nose-up posture change ($> 1°$), with bootstrapped 95% CI. (D,E) Largest magnitude slopes of the nose-down (D) and nose-up (E) best-fit sigmoids to data in (A), with bootstrapped 95% CI.
DOI: https://doi.org/10.7554/eLife.45839.015

Climbing mechanics are well-established for adult fishes (*Aleyev, 1977*; *Webb, 1994*; *Drucker and Lauder, 2002*). While we define a role for the pectoral fins in larval zebrafish climbing, the relevant kinematics remain unknown. Our work establishes several important constraints on the maturation of pectoral fin function. First, fin loss had no apparent impact on the ability of larvae to rotate their bodies in the pitch axis. Consistently, we observed that pectoral fins were located rostro-caudally near the estimated body center of mass, yielding a small moment arm in the pitch axis (*Drucker and Lauder, 2002*). Second, across development, larvae exhibited similar maximal attack angles, suggesting that lift (and thrust; *van Leeuwen et al., 2015*) forces scale with body mass as larvae develop. Comparable function of the pectoral fins with age may reflect their musculoskeletal simplicity in larvae (*Thorsen and Hale, 2005*; *Hale, 2014*). In contrast to larvae, mature fish use their pectoral fins both to steer and as proprioceptors (*Williams et al., 2013*; *Aiello et al., 2017*). Future

work relating pectoral fin kinematics to vertical wake structure in developing zebrafish stands to illuminate how morphological maturation permits increasingly sophisticated movements with age.

We found that larval zebrafish coordinated their pectoral fins and bodies, controlling them independently but using them synergistically to facilitate climbing. Importantly, larvae missing their utricular otoliths, that is, gravity-blind mutants (*Roberts et al., 2017*), did not coordinate fin and body actions despite performing each independent action typically. Two important conclusions follow from the mutant experiments. First, coordination of fin and body actions reflects patterned control, distinct from movements that are correlated simply due to biomechanics (*Collins et al., 2001*). Second, though gravity-blind mutants could swim with a normal dorsal-up orientation in the light, utricular information is necessary for properly coordinated climbing. In mutant fish, posture changes and attack angles (reflecting body- and fin-mediated climbing, respectively) were normal, but unrelated. Synergistic fin and body actions therefore reflect a neural transformation of vestibular information into correlated commands for climbing.

Our discovery that loss of the utricular otoliths abolishes synergistic fin and body actions reveals a sensory, and specifically vestibular, origin for the signals guiding coordinated climbing. On land, animals can infer their orientation relative to gravity from sensed pressure and muscle tension, allowing touch and proprioception to guide posture and locomotion (*Proske and Gandevia, 2012*; *Tuthill and Wilson, 2016*). In zebrafish, recent work has identified a class of spinal proprioceptors that provide feedback during axial locomotion (*Knafo and Wyart, 2018*), and ascending feedback from the spinal cord in swimming tadpoles can drive compensatory ocular reflexes (*Combes et al., 2008*). However, under water, the homogeneous physical environment necessitates vestibular strategies to guide coordinated locomotion with respect to gravity – such as the climbs we have studied here. Links between the vestibular system and postural orientation in the pitch axis are present in evolutionarily ancient vertebrates such as lamprey (*Orlovsky et al., 1992*). Vestibular information can drive pectoral fin movements in chondrichthyes (*Timerick et al., 1990*), one of the earliest classes in which pectoral fins appear (*Coates, 1994*). Considerable morphological (*Shubin et al., 2006*) and molecular (*Jung et al., 2018*) work underscores the importance of the pectoral fins in the evolution of terrestrial appendages and gaits necessary for locomotion. Our findings extend this work by linking sensed gravity to the underwater climbing behaviors these ancient appendages serve.

Considering the development of coordination an optimization process, maturation may be driven by a gradual approach to a fixed, optimal coordination, or a change in the definition of optimal coordination (*Sporns and Edelman, 1993*; *Adolph and Tamis-LeMonda, 2014*). Specifically, fin bias may increase with age because optimal fin use gradually becomes possible or because large fin bias becomes optimal. Based on simulations of swimming using empirical coordination at different ages (Appendix), we propose that coordination may develop due to a change in the definition of optimal locomotion. Specifically, younger larvae generate fin-body coordination that primarily minimizes the effort required to steer, while older larvae generate fin-dominated steering that is optimized more for maintaining balance. Ultimately, defining the contribution of cost function dynamics to the development of motor control and specifically coordination will require acute manipulations of performance or feedback.

The cerebellum has long been recognized for its role both in enabling (*Morton and Bastian, 2004*) and learning (*Thach, 1998*; *Bastian, 2006*) coordinated movements, though the computations responsible remain contentious (*Manto et al., 2012*). We found that ablation of cerebellar Purkinje cells perturbed fin-body coordination, leading to the production of conflicting actions in which larvae generated fin-mediated lift while making nose-down rotations. Furthermore, ablation changed fin-body coordination during nose-up rotations, causing larvae to pair stronger fin actions with the same body rotation. We conclude that the cerebellum acts to suppress lift generation by the fins during body rotations, and thereby prevents the production of conflicting actions. Purkinje cells in the lateral cerebellum of zebrafish, labeled in the driver line used here (*Takeuchi et al., 2015*), project to vestibular nuclei (*Matsui et al., 2014*) and respond to rotational visual stimuli (*Knogler et al., 2019*) and vestibular stimulation (*Favre-Bulle et al., 2018*; *Migault et al., 2018*). Intriguingly, cerebellar lesions in the dogfish result in profound impairment of pectoral fin reflexes (*Paul and Roberts, 1979*). Combining quantitative measurements of locomotion and molecularly-targeted perturbations has begun to yield new insights into cerebellar function (*Machado et al., 2015*). Similarly, climbing in zebrafish will likely prove to be a uniquely tractable entry point into the study of the cerebellum's role in the development of coordinated locomotion.

Our data are consistent with the hypothesis that the bottleneck to developing coordination lies in perceptual rather than motor capacity. Young larvae were physically capable of producing large attack angles with the fins while rotating their bodies to climb. Further, the range of body rotations larvae produced did not change across development. We therefore propose that swimming development does not require unlocking or composing new actions, but instead involves selecting a particular combination of equally functional innate actions (*Grillner and Wallén, 2004*; *Sporns and Edelman, 1993*). As in other vertebrates (*Beraneck et al., 2014*), the capacity of the vestibular system to stabilize gaze (*Bianco et al., 2012*) and posture (*Ehrlich and Schoppik, 2017a*) improves markedly with age. In mature animals, vestibular information is thought to be weighted by reliability for perceptual computations (*Angelaki et al., 2009*), consistent with learning rules (*Körding and Wolpert, 2004*) that may underlie locomotor development. We propose that the fundamental limit on locomotor development reflects not motor capabilities, but peripheral or central limits to perceived posture.

Existing literature suggests a candidate neural substrate for the vestibular signals that underlie perceived posture and promote coordination. The utricles transduce body orientation and self-motion but are insensitive to vertical forces orthogonal to the utricular macula (*Flock, 1964*; *Yu et al., 2012*), and should therefore be irrelevant for execution or perception of lift forces directly. A central origin for the signals that guide coordination is therefore more plausible, specifically in the utricle-recipient hindbrain vestibular nuclei (*Schoppik et al., 2017*; *Bagnall and Schoppik, 2018*). One of these, the tangential nucleus, contains 'Ascending-Descending' neurons (*Bianco et al., 2012*). These neurons are distinguished by bifurcating axons that project rostrally, ascending to a midbrain nucleus, the nucleus of the medial longidutinal fasciculus, a region responsible for descending control of swim kinematics (*Severi et al., 2014*; *Thiele et al., 2014*; *Wang and McLean, 2014*). Ascending-Descending neurons are anatomically poised to also relay otolith-derived signals to the pectoral fins, as they make descending projections to the locus of the pectoral fin motoneurons: the caudal hindbrain/rostral spinal cord (*Ma et al., 2010*). Pectoral fin motoneurons have been studied in the context of axial swimming, and this work has established that rostral hindbrain-derived signals are important for proper pectoral fin control during fast swimming (*Green and Hale, 2012*). In order to convey feedback to pectoral motoneurons about pitch-axis postural changes, Ascending-Descending neurons would need to encode angular velocity, consistent with the transient responses to pitch-axis posture changes of neurons in the fish tangential nucleus (*Suwa et al., 1999*) and with hindbrain vestibular responses more broadly (*Laurens et al., 2017*). We propose that the Ascending-Descending neurons in the tangential vestibular nucleus are therefore a candidate cellular substrate by which utricular information comes to regulate fin-body coordination.

Considerable effort has gone into defining the fundamental principles by which coordination emerges during locomotor development (*Bernstein, 1967*; *Newell and McDonald, 1994*). Maturation of coordination is thought to permit optimization of locomotion based on experience, and to facilitate adaptations to changing motor goals (*Sporns and Edelman, 1993*; *Thelen, 1995*; *Adolph, 1997*). Further, mature patterns of locomotion may be generally disfavored until balance can be maintained (*Thelen, 1984*; *Woollacott et al., 1989*; *Yogev-Seligmann et al., 2012*; *Adolph, 2016*). Broadly, the complexity of terrestrial biomechanics has made it difficult to understand why animals change the way they locomote, and how they accomplish these changes. We studied a simpler system – climbing underwater – to disentangle corporeal development from maturation of motor control. We discovered that the vestibular system shapes synergies between fin and body actions as larval fish learn to climb. Taken together, our work supports a fundamental role for the vestibular sense in the development of coordinated locomotion.

## Materials and methods

### Fish husbandry and lines

Procedures involving larval zebrafish (*Danio rerio*) were approved by the Institutional Animal Care and Use Committee of New York University. Fertilized eggs were collected from in-crosses of a breeding population of Schoppik lab wild-type zebrafish maintained at 28.5°C on a standard 14/10 hr light/dark cycle. Before 5 dpf, larvae were maintained at densities of 20–50 larvae per petri dish of 10 cm diameter, filled with 25–40 mL E3 with 0.5 ppm methylene blue. Subsequently, larvae were

maintained on system water in 2 L tanks at densities of 6–10 per tank and fed twice daily. Larvae received powdered food (Otohime A, Reed Mariculture, Campbell, CA) until 13 dpf and brine shrimp thereafter. Larvae were checked visually for swim bladder inflation before all behavioral measurements.

Transgenic fish with loss-of-function mutation of the inner ear-restricted gene, otogelin (otogelin -/-), exhibit delayed development of utricular otoliths (rock solo[AN66](Whitfield et al., 1996), RRID: ZDB-ALT-130411-212). Homozygous offspring were visually identified by lack of utricular otoliths at 2 dpf, and confirmed to have typical posterior position and morphology of saccular otoliths. For behavioral comparison siblings with unaffected otoliths were used.

Transgenic fish expressing KillerRed in Purkinje cells were generated using the Tg(aldoca: GAL4FF) line (Takeuchi et al., 2015), by crossing to Tg(UAS:KillerRed) (Del Bene et al., 2010).

## Morphological measurement
Dorsal-perspective, bright-field photomicrographs of 15 wild-type larvae across three clutches were taken at each developmental time-point using an eight megapixel iSight camera (Apple) through the ocular of a stereoscope (M80, Leica Microsystems). Larvae were immobilized dorsal up in 2% low-melting temperature agar (Thermo Fisher Scientific 16520). Body length and rostrocaudal position of the pectoral fin base were measured in Fiji (Schindelin et al., 2012) and compared to previously published estimates of center of mass (COM), estimated by modeling bodies as series of elliptic cylinders (Ehrlich and Schoppik, 2017a). Additionally, body and pectoral fin lengths were measured from 15 otogelin -/- larvae and 15 phenotypic controls (otog+/- or otog+/+, differentiated by absence of utricles) at 1 wpf.

## Surgery
Pectoral fins were amputated from wild-type larvae anesthetized in 0.02% ethyl-3-aminobenzoic acid ethyl ester (MESAB, Sigma-Aldrich E10521, St. Louis, MO). Pairs of anesthetized, body length-matched siblings were immobilized dorsal-up in 2% low-melting temperature agar (Thermo Fisher Scientific 16520), and both pectoral fins of one larva were removed by pulling the base of the fin at the scapulocoracoid laterally with forceps. Larvae were randomly allocated into groups without blinding. After amputation, both siblings were freed from the agar with a scalpel and allowed to recover in E3 for 4–5 hr prior to behavioral measurement.

## Cerebellar lesion
Cerebellar Purkinje cells were lesioned at 6 dpf specifically using transgenic expression of the photo-sensitizer, KillerRed. Larvae were mounted dorsal-up in agarose and anesthetized in MESAB. Control, transgenic larvae were anesthetized in MESAB in parallel. Larvae were randomly allocated into groups without blinding. Illumination conditions on a widefield microscope (Axio Imager M1, Zeiss, Oberkochen, Germany) were set under blue light (480/30 excitation filter from filter set 19002, Chroma Technology, VT) to visualize but not activate KillerRed. Light was focused through a 40x, 0.75NA water immersion objective (Zeiss Achroplan), stopped down to fill a 0.3 mm diameter region, and focused on the Purkinje cell somata. Green light (540/25 excitation filter from filter set 19004, Chroma Technology, VT) was then applied for 15 min, quenching KillerRed fluorescence. The power at the sample plane, measured at 540 nm with a 9.5 mm aperture silicon photodiode (PM100D power meter, S130C sensor, Thorlabs, NJ) was 14 mW. Fish were allowed to recover for 16–24 hr before behavioral measurements.

## Behavior measurement
Experiments were performed on 4 clutches of 32 wild-type siblings, with eight larvae per clutch recorded at 4 dpf and 1, 2, and 3 wpf as in a previous study (Ehrlich and Schoppik, 2017a). Additionally, 12 clutches of 12–16 larvae each were divided evenly and compared with and without amputation of the pectoral fins, both at 1 and 3 wpf (6 clutches each). Five clutches of 16 siblings each, eight lacking utricles (otogelin -/-) and eight phenotypic controls (otog+/- or otog+/+), were measured at 1 wpf before homozygous mutants develop utricles (Bagnall and Schoppik, 2018). Finally, 16 Tg(aldoca:GAL4FF;);Tg(UAS:KillerRed) siblings, 10 lesioned and 6 controls, were measured at 1 wpf in constant darkness.

Larvae were filmed in groups of 4–8 siblings in a glass tank (93/G/10 55 × 55×10 mm, Starna Cells, Inc, Atascadero, CA) filled with 24–26 mL E3 and recorded for 48 hr, with E3 refilled after 24 hr. The thin tank (10 mm) restricted swimming near the focal plane. Water temperature was maintained at 26 °C in an enclosure with overhead LEDs on a 14/10 hr light/dark cycle. Video was captured using digital CMOS cameras (BFLY-PGE-23S6M, Point Grey Research, Richmond, BC, Canada) equipped with close-focusing, manual zoom lenses (18–108 mm Macro Zoom 7000 Lens, Navitar, Inc, Rochester, NY) with f-stop set to 16 to maximize depth of focus. The field-of-view, approximately 2 × 2 cm, was aligned concentrically with the tank face. A 5W 940 nm infrared LED backlight (eBay) was transmitted through an aspheric condenser lens with a diffuser (ACL5040-DG15-B, Thor-Labs, NJ). An infrared filter (43–953, Edmund Optics, NJ) was placed in the light path before the imaging lens.

Video acquisition was performed as previously (*Ehrlich and Schoppik, 2017a*). Digital video was recorded at 40 Hz with an exposure time of 1 ms. To extract kinematic data online using the NI-IMAQ vision acquisition environment of LabVIEW (National Instruments Corporation, Austin, TX), background images were subtracted from live video, intensity thresholding and particle detection were applied, and age-specific exclusion criteria for particle maximum Feret diameter (the greatest distance between two parallel planes restricting the particle) were used to identify larvae in each image (*Ehrlich and Schoppik, 2017a*). In each frame, the position of the visual center of mass and posture (body orientation in the pitch, or nose-up/down, axis) were collected. Posture was defined as the orientation, relative to horizontal, of the line passing through the visual centroid that minimizes the visual moment of inertia, such that a larva with posture zero has its longitudinal axis horizontal.

Supplemental videos at high spatial resolution were alternatively filmed in a thinner glass tank (96/G/5 24 × 5 × 5 mm, Starna Cells, Inc) using a Sony IMX174 CMOS chip (ace acA1920-155um, Basler AG, Germany) equipped with a high-magnification fixed focus lens (Infinistix 0.5x, Infinity Optical Company, Boulder CO) and a high-pass infrared filter (Optcast 43948, Edmund Optics). Infrared illumination was provided by multiple high-power LEDs (5W 940 nm center wavelength, eBay); one, mounted behind the tank, provided transmitted light, passed through an aspheric condenser lens with diffuser (ACL5040-DG15-B) and a piece of Cinegel #3026 Filter paper (Rosco USA, Stamford CT). Three additional infrared LEDs were mounted between the lens and the tank to provide reflected illumination: one coupled to a fiber optic ring light (Optcast 54176, Edmund Optics) mounted on the lens barrel, and two additional bare LEDs mounted on either side of the tank at 45° angles. Full-frame (1900x1200, 8-bit) video capture was triggered at 60 Hz with a 7 ms exposure time.

## Behavior analysis

Data analysis was performed using Matlab (MathWorks, Natick, MA). Epochs of consecutively saved frames lasting at least 2.5 s were incorporated in subsequent analyses if they contained only one larva. Data were analyzed from the light phase during the first 24 hr of measurement, but excluded a 2 hr period following transition from the dark phase to minimize influence of light onset.

Deviation from horizontal was computed as the mean of absolute value of all postures observed. Instantaneous differences of body particle centroid position across frames were used to calculate speed. As previously (*Ehrlich and Schoppik, 2017a*), bouts were defined as periods with speeds exceeding 5 mm·s$^{-1}$, and consecutively detected bouts faster than $13\frac{1}{3}$ Hz were merged.

Numerous properties of swim bouts were measured or calculated. The maximum speed of a bout was determined from the largest displacement across two frames during the bout. The trajectory of a bout was defined as the direction of instantaneous movement across those two frames. Bouts with backwards trajectories (> 90°or < −90°, fewer than 1% of bouts across all ages) were excluded from analysis. The displacement across each pair of frames at speeds above 5 mm·s$^{-1}$ were summed to find net bout displacement. Attack angle was defined as the difference between trajectory and posture of a larva at the peak speed of a bout, such that a larvae pointed horizontally and moving vertically upwards had an attack angle of 90°. Posture change during a bout was defined as the difference in body orientation observed 25 and 75 ms before peak speed, when rotations correlate with changes to trajectory (*Ehrlich and Schoppik, 2017b*).

Instantaneous bout rate was defined as the inverse of the interval between the first frame exceeding 5 mm·s$^{-1}$ in each of two successive bouts captured in a single epoch. Durations of bouts were calculated by linearly interpolating times crossing 5 mm·s$^{-1}$ on the rising and falling phases of each bout. Inter-bout duration was computed as the difference between inverse bout rate (instantaneous bout period) and bout duration. Vertical displacement during an inter-bout was computed as the difference between the vertical position of larva centroid at the end and start of each inter-bout.

A logistic function was used to fit the sigmoidal relationship between attack angle ($\gamma$) and posture change ($r$), based on a simple formulation,

$$\gamma(r) = \gamma_0 + \frac{\gamma_{max}}{1 + e^{-k(r-r_0)}},$$ (1)

in which $\gamma_0$ gives the most negative attack angle (on average, in deg), ($\gamma_{max} + \gamma_0$) gives the largest positive attack angle (on average, in deg), and $k$ is the steepness parameter (in deg$^{-1}$). From the derivative of *Equation (1)*, sigmoid maximal slope (dimensionless, found at $r = r_0$) is given by $k\gamma_{max}$ / 4. Because empirical data at all ages rose from the lower asymptote at similar values of posture change, sigmoid center position ($r_0$) was itself defined from a parameter for rise position ($r_{rise}$, posture change at which the sigmoid rises to 1/8 of its upper asymptote):

$$r_0 = \frac{kr_{rise} + log(\frac{-\gamma_0 - 7\gamma_{max}}{\gamma_0 + \gamma_{max}})}{k}.$$ (2)

Parameter fits and confidence intervals were estimated in Matlab using a nonlinear regression-based solver (Levenberg-Marquardt) to minimize the sum of squared error between empirical and estimated attack angles given empirical posture changes. Initial parameter values were $k$ = 1 deg$^{-1}$, $\gamma_0$=−0.2°, $\gamma_{max}$=20°, and $r_{rise}$ = −1°. Data were pooled across all bouts in a given group (age or utricle phenotype). To fit data from individual clutches, pools of available swim bouts were increased by including data from 48 hr of swimming, rather than 24 hr. Given that $\gamma_0$, $\gamma_{max}$, and $r_{rise}$ exhibited no consistent or significant trend with age (*Table 1*), values were fixed at means across all ages (−2.97 °, 17.02 °, and −1.11°, respectively) and sigmoid steepness was evaluated. One-parameter sigmoids fit empirical data well across development relative to four-parameter sigmoids (*Table 1*).

In contrast, a one-parameter sigmoid poorly fit empirical data for *otogelin - /-* larvae ($R^2 = -0.17$), which had uncorrelated attack angles and posture changes. Freeing the $r_{rise}$ parameter gave a sigmoid with a steepness of approximately 0 that fit slightly better than mean attack angle ($R^2 = 0.005$), so fin biases for *otogelin - /-* and control larvae were calculated from maximal slopes of two-parameter sigmoids.

From sigmoid fits, empirical fin bias ($\hat{\alpha}$) was computed as an index of maximal sigmoid slope (slope/(1+slope)). Fin bias therefore reflected the ratio of attack angle to posture change in a given climb. For sigmoids with positive steepness ($k$),

$$\hat{\alpha} = \frac{k \cdot 4.25°}{1 + k \cdot 4.25°}.$$ (3)

In the generative swimming model (Appendix), commands to the fins (to generate attack angle) and body (to produce a posture change) were both calculated using the fin bias parameter ($0 \leq \alpha \leq 1$), such that attack angle and posture change had a maximal ratio of $\alpha/(1-\alpha)$. In this way, fin bias reflects the ratio of attack angle and posture change for both empirical and simulated swimming.

A single sigmoid (*Equation 1*) poorly fit empirical data for larvae with cerebellar lesions ($R^2 = 0.100$) but not controls (0.175). While the single sigmoid accurately fit data with positive posture changes and identified significant differences in steepness across conditions, it failed to capture the tendency in lesioned animals to pair positive attack angles with negative posture changes. Instead, these data were fit with the sum of two sigmoids, one reflected about the vertical axis:

$$\gamma_p(r) = \gamma(r) + \frac{\chi\gamma_{max}}{1 + e^{k(r+r_0)}}.$$ (4)

The relative amplitudes of the two sigmoids were scaled by parameter $\chi$, and the nose-up sigmoid amplitude was defined as 17.02° as for the one-parameter sigmoid. This four parameter

function was fit from initial values of $\chi = 0.5$, $k = 1$ deg$^{-1}$, $\gamma_0 = -5°$, and $r_{rise} = 0°$. For control larvae, the $\chi$ parameter did not significantly differ from zero ($0.13 \pm 0.36$), yielding a negligible contribution from the reflected sigmoid (see Results). Compared to the one-sigmoid function, the two-sigmoid function had minor effects on the goodness-of-fit and solutions for control data ($R^2 = 0.178$; $k = 0.36$ vs. $0.40$). In contrast, for lesion data the two-sigmoid function improved goodness-of-fit ($R^2 = 0.137$), yielded a value for $\chi$ that significantly differed from zero ($0.63 \pm 0.25$), and drastically increased sigmoid steepness ($k = 0.59$ vs. $1.53$ deg$^{-1}$).

## Statistics

Significance level was defined at 0.05. Pairwise t-tests were used to assess the effects of fin amputation on swim properties from sibling groups at both 1 and 3 wpf. Morphological properties were analyzed by one-way ANOVA assuming independence of all individual larvae. Two-way ANOVA with factors of age and clutch were used to assess effects on swim properties from larvae 1, 2, and 3 wpf, with significant main effects of age followed by Tukey's post-hoc tests. One exception was the coefficient of determination of trajectory and posture, which failed the assumption of homoscedasticity; effect of age was assessed with a non-parametric Kruskal-Wallis test. Sample sizes were defined without power analysis based on previous studies using the same behavioral apparatus (*Ehrlich and Schoppik, 2017a*).

## A technical note on terminology

We use the term 'attack angle' to describe the difference between the orientation of the body's long axis and the trajectory of swimming. As our fish swim in stagnant water, this trajectory is assumed to oppose the direction of flow. Our definition describes the orientation of an element's long axis relative to flow, consistent with the terminology in fluid dynamics. Our fish vary the direction of motion with respect to the body, and we are specifically interested in control of steering. Accordingly, we consider attack angles of the body because they are the consequence of forces orthogonal to the body long axis – by which the fish steer upwards and downwards. We refer to these upwards forces as 'lift' and attribute them to pectoral fins by inference, based on loss of positive attack angles following fin amputation. However, we have no data that speak to fin kinematics or the mechanics of force production; for example whether lift is generated by fin strokes or flow over fins due to body-mediated motion, or if the fins produce vertical thrust (*Aleyev, 1977*; *Westneat et al., 2004*).

## Data sharing

Raw data and analysis code are available at http://www.schoppiklab.com/.

## Acknowledgements

Research was supported by the National Institute on Deafness and Communication Disorders under award DC012775 to DS and an Emerging Research Grant from Hearing Health Foundation to DE. The authors would like to thank Martha Bagnall for sharing the *otogelin - /-* line and Emre Aksay for sharing the *aldoca:GFF* line. The authors would like to thank Başak Sevinç, Katherine Harmon, Marie Greaney, Tim Gerson, and Shane Hunt for assistance with animal husbandry; Simon D Sun for apparatus construction; Katherine Nagel, Kishore Kuchibhotla, Sam McKenzie, Timothy Currier, Kristen D'Elia, Kyla Hamling, Dena Goldblatt and members of the Schoppik and Nagel labs for helpful comments.

## Additional information

### Funding

| Funder | Grant reference number | Author |
| --- | --- | --- |
| National Institute on Deafness and Other Communication Disorders | DC012775 | David Schoppik |

| Hearing Health Foundation | | David E Ehrlich |
| National Institute on Deafness and Other Communication Disorders | DC016316 | David Schoppik |

The funders had no role in study design, data collection and interpretation, or the decision to submit the work for publication.

### Author contributions
David E Ehrlich, Conceptualization, Funding acquisition, Investigation, Visualization, Methodology, Writing—original draft, Writing—review and editing; David Schoppik, Conceptualization, Supervision, Funding acquisition, Methodology, Writing—review and editing

### Author ORCIDs
David E Ehrlich  https://orcid.org/0000-0003-2823-731X
David Schoppik  https://orcid.org/0000-0001-7969-9632

### Ethics
Animal experimentation: This study was performed in strict accordance with the recommendations in the Guide for the Care and Use of Laboratory Animals of the National Institutes of Health. Procedures involving larval zebrafish (*Danio rerio*) were approved by the Institutional Animal Care and Use Committee of New York University (protocol #16-00561).

### Decision letter and Author response
Decision letter https://doi.org/10.7554/eLife.45839.024
Author response https://doi.org/10.7554/eLife.45839.025

## Additional files

### Supplementary files
• Transparent reporting form
DOI: https://doi.org/10.7554/eLife.45839.016

### Data availability
Raw data and analysis code are available on Dryad (http://doi.org/10.5061/dryad.j9kd51c7d).

The following dataset was generated:

| Author(s) | Year | Dataset title | Dataset URL | Database and Identifier |
| --- | --- | --- | --- | --- |
| Ehrlich DE | 2019 | Data from: A primal role for the vestibular sense in the development of coordinated locomotion | http://doi.org/10.5061/dryad.j9kd51c7d | Dryad, 10.5061/dryad.j9kd51c7d |

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

## Appendix 1

DOI: https://doi.org/10.7554/eLife.45839.017

### Evaluating optimization rules with a generative model of fin-body coordination

Why do developing larvae change how they divide labor between the fins and body? In other words, what cost function are larvae optimizing when they regulate fin-body coordination? To address this question, we built a simple computational model that allowed us to parameterize the division of labor between the fins and body (*Appendix 1—figure 1A*, details in Materials and methods). In this control-theoretic model, a larva swam towards a target (up or down) by comparing the target's direction to the direction it would swim without steering (its current posture). From this difference the larva generated a steering command. The larva steered its swim bouts by using its fins to generate lift and its body to direct thrust.

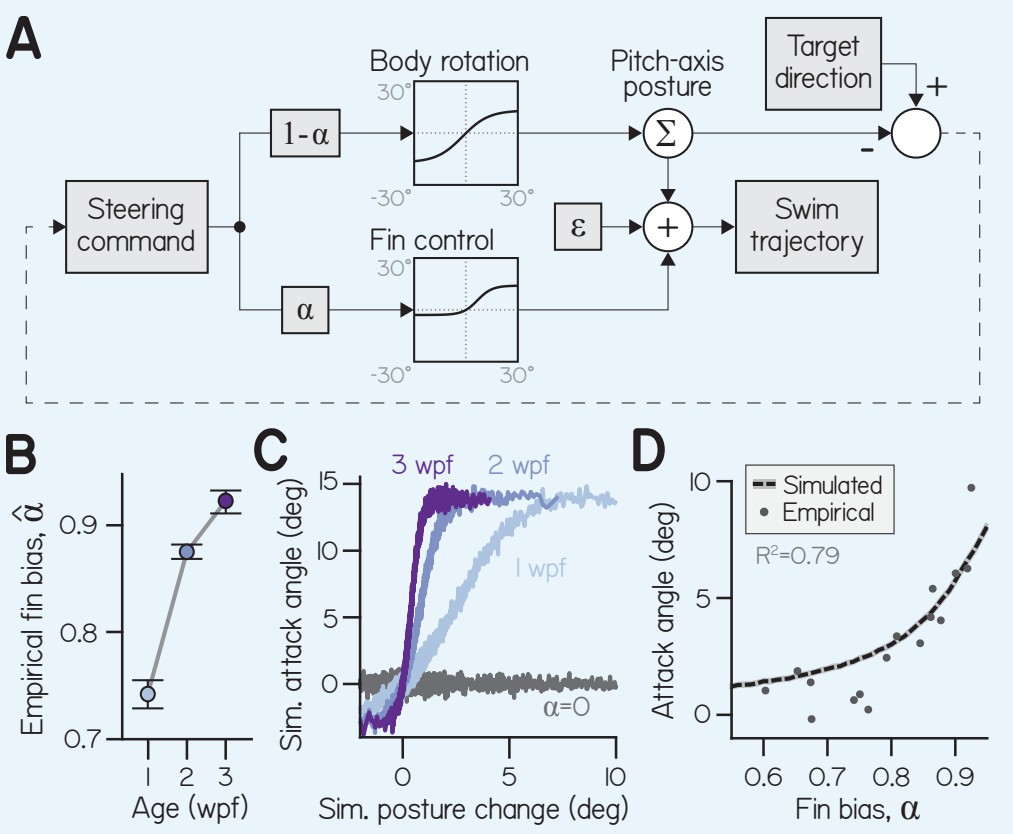

**Appendix 1—figure 1.** A one-parameter control system captures fin-body coordination *in silico*. (**A**) Circuit diagram to transform pitch-axis steering commands into climbing swims using the body and pectoral fins. Steering commands are defined by the direction of a target in egocentric coordinates. The relative weight of commands to rotate the body (to direct thrust) and produce an attack angle with the fins (by generating lift) is dictated by fin bias ($\alpha$). To model physical transformations from commands into kinematic variables, commands to the body and fins are filtered to impose empirically-derived ceilings and floors on posture changes and attack angles (see Materials and methods). Swim trajectory is defined by posture (fish propel where they point) but modified by attack angle and error ($\varepsilon$). (**B**) Empirical fin bias ($\hat{\alpha}$), computed from maximal sigmoid slope (slope/(1+slope)), as a function of age with 95% confidence intervals. (**C**) Attack angle as a function of posture change, plotted as means of equally-sized bins. Climbs to 100,000 targets were simulated using empirical fin bias ($\hat{\alpha}$) from

1, 2, and 3 wpf larvae, and at $\alpha = 0$ for comparison. (**D**) Mean attack angle for simulated larvae with parameterized fin bias (line), superimposed on empirical attack angles and fin biases ($\hat{\alpha}$) for each clutch at each age. Simulated attack angles at $\hat{\alpha}$ account for 79% of variation in empirically observed attack angles ($R^2$).

DOI: https://doi.org/10.7554/eLife.45839.018

Simulated larvae coordinated their fins and bodies by controlling both effectors with a mutual command. To vary the ratio of fin and body actions (attack angles and posture changes, respectively), the command was differentially scaled for the fins and body. Commands to the fins were weighted by a fin bias parameter ($0 \leq \alpha \leq 1$) and commands to the body by ($1 - \alpha$). Effector-specific commands were therefore positively correlated (when $\alpha / = 0$ and $\alpha \neq 1$) in a ratio equal to $\alpha/(1-\alpha)$. Given this formulation, we could infer empirical fin biases ($\hat{\alpha}$) from the ratio of empirical attack angles and posture changes, given by sigmoid slope ($\hat{\alpha}$ = slope/(1+slope); *Equation 3*). Empirical fin bias increased significantly with age (from 0.74 at 1 wpf to 0.92 at 3 wpf) like sigmoid slope, but ranged from 0 to 1 (*Appendix 1—figure 1B*; *Table 1*). Commands were transformed into kinematic variables according to physical transfer functions (*Appendix 1—figure 1A*) that increased approximately linearly near the origin, such that weak commands were faithfully transformed to movement; for large positive and negative commands, transfer functions reached asymptotes to model physical limitations. The asymptotes imposed empirically-derived constraints on the range of posture changes (−17.0 to +13.2°) and attack angles (−2.9 to +14.0°) of each bout. Additionally, Gaussian noise was added to swim trajectory to model errors in motor control and effects of external forces like convective water currents that move larvae ($\varepsilon$).

The model permitted simulation of larvae across development, because sigmoid slope (and therefore fin bias) captured developmental changes to swimming. We simulated larvae from each age group (100,000) identically, save for age-specific $\hat{\alpha}$, as they climbed in series of bouts until reaching targets positioned half the tank away (25 mm). We placed the targets in directions randomly drawn from observed climbing trajectories (see Materials and methods). Simulated attack angles and posture changes were sigmoidally related, with steeper sigmoid slopes for older larvae (*Appendix 1—figure 1C*). Simply by varying fin bias, simulated larvae exhibited mean attack angles comparable to empirical values (*Appendix 1—figure 1D*). Simulated attack angles at age- and clutch-specific $\hat{\alpha}$ yielded close approximations of attack angle ($R^2 = 0.79$). A model with a single parameter that scales divergent commands can therefore produce fin-body coordination and mimic climbing behavior across development.

## Increasing fin bias improves balance but costs effort in silico

Body-mediated climbing requires orienting the body upwards, causing posture to deviate from horizontal. Orienting upwards requires an initial energetic investment, but once larvae acquire the proper trajectory they can cease further turning. By comparison, fin-mediated climbing causes no postural deviation from horizontal and requires continued investment throughout the climb. Because larvae combine body- and fin-mediated climbing according to fin bias, we measured the effect of fin bias on posture variation and climbing efficacy.

We parameterized fin bias in silico, simulating larvae that climbed solely by generating lift with their fins ($\alpha = 1$) or solely by changing body posture ($\alpha = 0$), the former yielding larvae that never deviated from horizontal (*Appendix 1—figure 2A*). As fin bias increased, larvae remained closer to horizontal while climbing. After five bouts towards the steepest drawn target (63°), larvae swimming without using their fins ($\alpha = 0$) deviated 52° from horizontal, larvae with small fin bias (like those at 1 wpf, $\hat{\alpha} = 0.74$) deviated 39°, and larvae with large fin bias (like those at 3 wpf, $\hat{\alpha} = 0.92$) deviated only 13°.

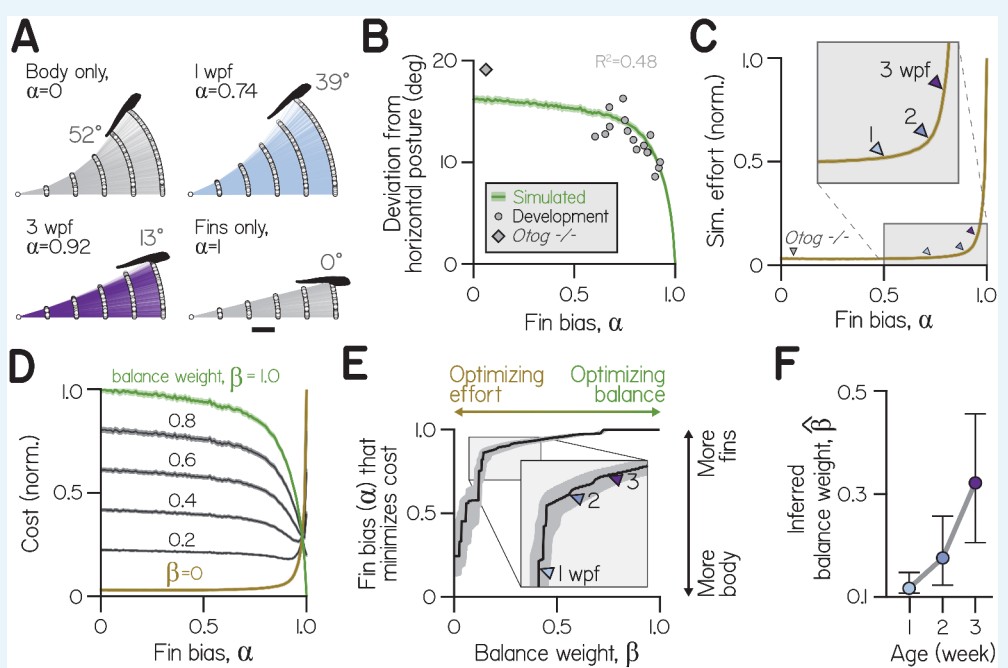

**Appendix 1—figure 2.** Effects of fin-body coordination on balance-effort trade-off. (**A**) Trajectories (lines) and initial positions (dots) of bouts simulated with the control system in (**A**) at fin biases of 0, 0.74 ($\hat{\alpha}$ at 1 wpf), 0.92 ($\hat{\alpha}$ at 3 wpf), and 1.0, for 1000 larvae swimming towards targets 25 $\mu$m away. Posture following the fifth bout of the steepest climb is superimposed. Scale bar equals 1 mm. (**B**) Simulated absolute deviation from horizontal posture as a function of $\alpha$, plotted as mean (green line) and bootstrapped 99% confidence intervals (shaded band). Data are superimposed on empirical values for individual clutches of a given age (circles, *Development*, $R^2 = 0.48$) and *otog-/-* larvae (diamond). (**C**) Effort, the sum of squared motor commands to the body and fins, from simulations in (**B**) normalized and plotted as a function of $\alpha$ as mean (line) and bootstrapped 99% confidence intervals (shaded band). Empirical fin biases at 1, 2, and 3 wpf and for *otog-/-* larvae are indicated with triangles. (**D**) Cost as a function of fin bias, computed as sums of normalized curves in (**B**) and (**C**) weighted by $\beta$ (balance weight) and $(1 - \beta)$, respectively (*left*). When $\beta = 1$ (green), cost is equivalent to normalized deviation from horizontal. When $\beta = 0$ (ochre), cost is equivalent to effort. Intermediate cost functions are plotted for $\beta$ increasing by 0.2, with 99% confidence interval (shaded band). (**E**) Fin bias at which cost was minimized is plotted at each value of balance weight, with 95% confidence intervals. (**F**) Inferred balance weight ($\hat{\beta}$) is plotted as a function of age, with 95% confidence intervals. This weight gives the cost function minimized by empirical fin bias at a given age (from the curve in E).

DOI: https://doi.org/10.7554/eLife.45839.019

We found that the relationship between posture variation and fin bias was similar for empirical and simulated larvae (*Appendix 1—figure 2B*). Larger fin biases were associated with smaller deviations from horizontal, reflecting better balance. Although the model was not explicitly fit to postural variables, simulated deviations from horizontal explained 48% of empirical variance for clutches and time-points across development (with fin biases spanning from 0.60 to 0.93). Additionally, at fin biases below 0.5, simulated deviations were consistent near 17°. *otog-/-* larvae exhibited very small fin bias ($\hat{\alpha} = 0.06$) and large deviations from horizontal similar to simulated values at low $\alpha$ (19.1°). We conclude that simulations accurately captured the consequences of fin bias for balance, with greater fin bias allowing larvae to remain nearer horizontal.

To quantify how fin bias influenced swimming efficacy when climbing to targets, we measured effort. In order to define effort, we made a simplifying assumption: that effort would scale monotonically with the size of the desired movement. To begin, we defined effort as the

sum of squared motor commands for steering. Our initial choice to make effort scale quadratically with the control signal follows the convention in *Todorov and Jordan (2002)*; *Guigon et al. (2007)*.

Swimming with greater reliance on the fins, with larger fin bias, made climbing more effortful. Larvae at lower fin biases climbed farther in the same number of swims, benefiting from the cumulative effects of body rotation on posture (*Appendix 1—figure 1A*). Simulated larvae at 1 wpf gained two-thirds more elevation (4.25 mm) than larvae at three wpf (2.55 mm) and nearly three times as much as larvae swimming solely with the fins ($\alpha = 1$, 1.54 mm).

Effort increased monotonically with fin bias (*Appendix 1—figure 2C*). Steering solely with the fins ($\alpha = 1$) required 32 times more effort, on average, than steering solely by rotating the body ($\alpha = 0$). Because larvae could ultimately achieve steeper trajectories when steering with the body, larvae often required more swims to reach a target at large fin biases. By simulating effort at empirical fin biases, we estimated that older larvae swam with greater effort; efforts at empirical fin biases (relative to effort at $\alpha = 1$) corresponded to 3.6%, 5.5%, and 7.9% at 1, 2, and 3 wpf, respectively (*Appendix 1—figure 2C*, triangles). Further, very small fin bias observed in *otog-/-* larvae approximately corresponded with the least effortful swimming (3.1% of effort at $\alpha = 1$). Effort also increased monotonically as a function of fin bias when computed as the sum of absolute motor commands as well as squared or absolute accelerations (*Appendix 1—figure 3*). We conclude that larvae achieve the least effortful climbing at low fin biases. Furthermore, as larvae develop and adopt larger fin biases, they swim with increasing effort.

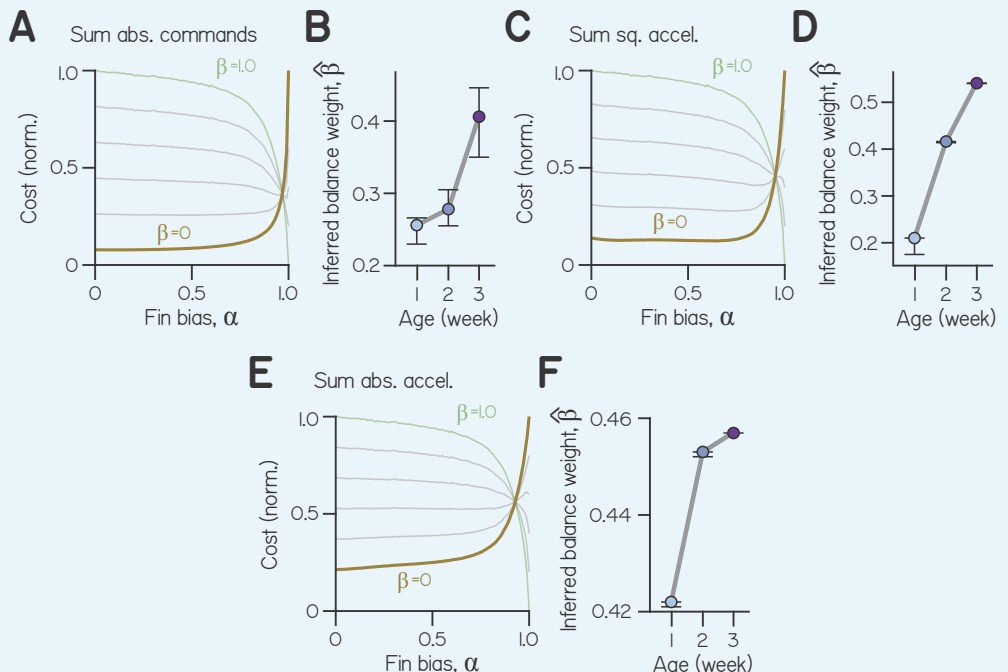

**Appendix 1—figure 3.** Steering cost functions computed from various formulations of effort. (**A**) Cost as a function of fin bias for $\beta$ (balance weights) of 0 (ochre, composed solely of effort), 0.2, 0.4, 0.6, 0.8, and 1 (green), for effort computed as the sum of absolute motor commands. (**B**) Inferred balance weight as a function of age, with bootstrapped 99% CI, for effort computed as the sum of absolute motor commands. (**C**) Cost as a function of fin bias for effort computed as the sum of squared accelerations. (**D**) Inferred balance weight as a function of age, with bootstrapped 99% CI, for effort computed as the sum of squared accelerations. (**E**) Cost as a function of fin bias for effort computed as the sum of absolute accelerations. (**F**) Inferred balance weight as a function of age, with bootstrapped 99% CI, for effort computed as the sum of absolute accelerations.

DOI: https://doi.org/10.7554/eLife.45839.020

Given that low fin biases required less effort and large fin biases facilitated balance, we next related the consequences of fin bias in each domain. We composed cost functions from terms for both balance and effort (*Appendix 1—figure 2D*). Specifically, we tested whether combinations of balance and effort terms could prescribe specific fin biases for optimal swimming. Cost functions are inherently dimensionless, so we summed normalized curves for balance (deviation from horizontal as a function of $\alpha$, *Appendix 1—figure 2B*) and effort (sum of square motor commands as a function of $\alpha$, *Appendix 1—figure 2C*). To vary the relative importance of balance and effort terms, we weighted them by $\beta$ (balance weight) and 1-$\beta$, respectively.

Our behavioral experiments established that, as they develop, larvae appear to coordinate their fins and bodies in such a way as to permit more balanced swimming (*Figure 2*). In our simulations, at each age larvae swam with a fin bias that minimized cost for a particular combination of balance and effort. Empirical fin biases minimized distinct cost functions composed from different balance weights (*Appendix 1—figure 2E*). From the cost functions that were minimized by empirical fin biases, we estimated the inferred balance weight ($\hat{\beta}$) at each age. Fin bias of larvae at 1 wpf minimized a cost function composed from a very low inferred balance weight (*Appendix 1—figure 2F*) $\hat{\beta} = 0.12$). Inferred balance weight increased by 2 wpf and significantly by 3 wpf, to 0.18 and 0.32, respectively. This framework is therefore consistent with our behavioral observations.

The specific values obtained for inferred balance weight reflect the choice to define 'effort' as the sum of the squared motor commands for steering. For comparison, we also computed effort as the sum of absolute motor commands, as well as the sum of squared or absolute accelerations (see Methods for simulating swimming). Inferred balance weight increased monotonically and significantly with age for all definitions of effort tested (*Appendix 1—figures 3B, 3D, 3F*). We parameterized β and found the optimal fin bias ($\alpha^*(\beta)$), the fin bias at which cost was minimized (*Figure 2E*, *Appendix 1—figures 3A, 3C, 3E*). As $\beta$ increased and cost functions grew more similar to deviation from horizontal, cost was minimized at larger fin biases. When $\beta = 0$ and the cost function solely reflected effort, the optimal fin bias was that which minimized effort ($\alpha^*(0) = 0.24$). Conversely, maximal fin bias was optimal for a range of cost functions that heavily weighted balance ($\beta > 0.72$). Our model can therefore generalize for different formulations of 'effort'.

By providing a framework to contextualize our observations, the model offers a way to understand the trade-offs facing developing larvae. We conclude that larvae regulate fin-body coordination to optimize balance and effort, and that development of fin-body coordination can be explained by an increase in the importance of balance relative to effort.

## Conclusions about optimizing coordination

We modeled the development of coordination as an adaptive process driven by dynamic optimization rules (*Kugler et al., 1980*; *Izawa et al., 2008*). To explore how larvae might implement coordination optimization rules, we quantified the effects of coordination parameters on performance variables, balance and effort (*Rigoux and Guigon, 2012*; *O'Sullivan et al., 2009*). We defined balance as deviation of posture from the horizontal, per our behavioral observation that fin-body synergies could minimize deviation from horizontal posture (*Figure 2A*). As the precise form of 'effort' is unknown, we assumed only that it would increase monotonically. We estimated effort either by scaling the control variables, as previously postulated in *Todorov and Jordan (2002)*; *Guigon et al. (2007)*, or by scaling kinematic variables proportional to the forces produced. We observed that steering with the body had a distinct advantage: rotations reoriented the body toward the target, minimizing the need for subsequent steering. In our model the more a larva used its pectoral fins to climb, the more effort (however defined) it expended to reach its target. However, consistent with our behavioral observations, steering with fins enabled climbing without changing trunk posture. We conclude that optimal fin bias reflects the relative importance of balancing and minimizing effort.

Taking our behavioral data and simulations together, we conclude that body-dominated climbing of young larvae is optimized primarily for effort, while fin-body synergies of older larvae are optimized both for effort and balance. Postural stability emerges as a key performance variable with development, so we propose that balance plays a primal role in the development of coordinated locomotion.

## Methods for simulating swimming

We made a generative swimming model in Matlab to estimate how fin bias impacts balance and effort while climbing. Simulated larvae moved in two dimensions (horizontal, $x$, and vertical, $z$) by making series of swim bouts ($b = 1, ..., n$) of variable trajectory ($t$) and fixed displacement (1.27 mm, the mean empirical displacement across all ages). Larvae swam from an origin at (0,0) such that the position after bout $b$ was determined by the trajectory of all preceding bouts. For the horizontal dimension, in mm:

$$x(b) = 1.27 \sum_{i=1}^{b} \cos(t(i)). \tag{1}$$

Larvae swam until traversing $\geq 99\%$ of both the horizontal and vertical distances from the origin to a target, located at distance $d$ and angle $\phi$ from the origin, or $(d \cdot \cos(\phi), d \cdot \sin(\phi))$.

Larvae could control $t$ during each bout through body rotation ($r(b)$) and by creating an attack angle with the pectoral fins ($\gamma(b)$). Body rotations allowed larvae to control their posture, which defined the direction of thrust. Larvae began swimming at horizontal posture (0°), meaning $\Theta$ during bout $b$ was given by the sum of rotations during that and all preceding bouts:

$$\Theta(b) = \sum_{i=1}^{b} r(i). \tag{2}$$

For each bout, trajectory was defined as the sum of posture ($\Theta(b)$), attack angle ($\gamma$), and a noise term ($\varepsilon$, defined below):

$$t(b) = \Theta(b) + \gamma(b) + \varepsilon. \tag{3}$$

To steer, larvae could directly vary $\gamma$ with the fins or influence $\Theta$ by rotating their bodies.

Movement noise ($\varepsilon$) was introduced to model motor errors and convective water currents that push larvae while they swim. Assuming finless larvae actively produce no attack angles ($\gamma = 0$; **Figure 1B and C**), their empirical attack angles reflect external forces ($\varepsilon = t - \Theta$). Therefore, simulated $\varepsilon$ for each bout was randomly drawn from a Gaussian distribution with a mean of 0 and standard deviation measured from empirical attack angles of finless larvae at 3 wpf (11.36°).

To make concerted posture changes and attack angles that steered toward a target, both $r$ and $\gamma$ were derived from a variable steering command ($c(b)$, in degrees) that provided feedback about the direction of the target. This command was defined before each bout and gave the direction of the target before the bout in egocentric terms (relative to the posture, $\Theta(b - 1)$, and position of the larva $(x(b - 1), z(b - 1))$). For a larva oriented toward the target, $c = 0$ such that no steering occurred. For the first bout, angle $c$ equaled $\phi$, and thereafter (for $b>1$)

$$c(b) = tan^{-1} \left( \frac{d \cdot sin(\phi) - z(b-1)}{d \cdot cos(\phi) - x(b-1)} \right) - \Theta(b-1). \tag{4}$$

Rather than swim upside-down, model larvae were assumed to make yaw-axis turns (side-to-side) to keep the target horizontally forwards; if a larva swam past the target, its horizontal position was simply reflected about the horizontal position of the target, such that $(d \cdot cos(\phi) - x)$ was always greater than 0.

Commands for attack angle ($\gamma'$) and body rotation ($r'$) were computed as complementary fractions that summed to the common steering command, $c$. The relative magnitude of $\gamma'$ and $r'$ was dictated by fin bias, $\alpha$ (defined from [0,1]), according to

$$\gamma'(b) = \alpha \cdot c(b) \qquad (5)$$

and

$$r'(b) = (1-\alpha) \cdot c(b). \qquad (6)$$

When $\alpha = 1$ larvae steered solely by generating attack angles with the fins, and when $\alpha = 0$ steered solely with posture changes. When $\alpha$ adopted intermediate values, the ratio of fin commands to body rotation commands was therefore $\alpha/(1-\alpha)$.

To transform commands ($\gamma'$ and $r'$) into kinematic variables ($\gamma$ and $r$), we modeled physical limitations as a ceiling and floor imposed with logistic functions. These physical transfer functions for the fins and body had maximal slopes of 1 and were constrained to the origin, faithfully transforming commands over a certain range but reaching asymptotes at positive and negative extremes (*Appendix 1—figure 1A*). The fin transfer function had asymptotes defined by empirical best-fit sigmoids to attack angle vs. posture change, averaged across ages (*Table 1*). The lower asymptote equaled $\gamma_0$ ($-2.94°$) and the upper asymptote equaled $\gamma_{max} + \gamma_0$ (14.04°). Given that the fin transfer function was also constrained to have maximal slope of 1 and pass through the origin, attack angle for a given bout was computed from the fin command according to

$$\gamma(\gamma'(b)) = -2.94° + \frac{16.98°}{1+e^{-k(\gamma'(b)-6.64°)}}, \qquad (7)$$

where $k = 0.24$ deg$^{-1}$. The body rotation transfer function was also constrained to have maximal slope of 1, pass through the origin, and have a range defined by the middle 99.9% of empirical body rotations (from $-16.98°$ to 13.15°). Body rotation for a given bout was computed from the rotation command according to

$$r(r'(b)) = -9.40° + \frac{17.58°}{1+e^{-k(r'(b)+1.92°)}}, \qquad (8)$$

where $k = 0.13$ deg$^{-1}$.

To assess correlations of $\gamma(b)$ and $r(b)$ at age-, phenotype-, and clutch-specific values of $\hat{\alpha}$, we simulated 100,000 larvae at each fin bias swimming to one target at $d=25$ mm (half the length of the empirical tank). The direction of the target, $\phi$, was randomly drawn from the positive lobe of a Gaussian distribution of mean 0 and standard deviation of 20.67° (that of trajectories of empirical bouts pooled across all ages). We also examined how deviation from horizontal, the mean of absolute value of simulated postures ($\Theta(b)$), as well as mean attack angle varied as a function of $\alpha$, parameterized from 0 to 1 in increments of 0.01. Given that simulated larvae could deviate widely from horizontal, we computed circular mean posture in Matlab using CircStat (*Berens, 2009*). After a simulated larva reached its target in $n$ bouts, effort ($E$) was computed as the sum of squared steering commands,

$$E = \sum_{i=1}^{n} c(i)^2. \qquad (9)$$

For comparison, effort was also calculated as the sum of absolute motor commands ($\Sigma|c(i)|$), as well as functions of acceleration. Given that simulated larvae had constant bout duration and displacement, ignoring drag makes angular acceleration proportional to body rotation and dorsal acceleration proportional to the sine of attack angle. We normalized these kinematic variables to their maximum values and summed the two to compute effort equivalent to the sum of squared accelerations

$$E = \sum_{i=1}^{n} \left(\frac{r(i)}{15.1°}\right)^2 + \left(\frac{\sin(\gamma(i))}{0.29}\right)^2 \tag{10}$$

and the sum of absolute accelerations

$$E = \sum_{i=1}^{n} \left|\frac{r(i)}{15.1°}\right| + \left|\frac{\sin(\gamma(i))}{0.29}\right|. \tag{11}$$

Bootstrapped confidence intervals were calculated by resampling simulated larvae 1000 times with replacement.

## Cost function derivation

Cost ($Q(\alpha)$) was calculated as a weighted sum of normalized deviation from horizontal ($\Theta^*(\alpha)$, *Appendix 1—figure 2B*) and normalized effort ($E(\alpha)$, *Appendix 1—figure 2C*), after both were interpolated fivefold and smoothed with a 25 point sliding window. Deviation from horizontal was scaled by a balance weight coefficient ($0 \leq \beta \leq 1$) and effort was scaled by ($1-\beta$), such that

$$Q(\alpha) = \beta\, \Theta^*(\alpha) + (1-\beta)E(\alpha). \tag{12}$$

Parameterizing $\beta$ yielded a family of cost functions. Finding the fin bias at which cost was minimized gave the optimal fin bias, $\alpha^*(\beta)$. Confidence intervals on optimal fin bias were taken as the farthest neighboring values of $\beta$, larger and smaller, at which the bootstrapped 2.5 percentile of cost exceeded the minimal cost. Inferred balance weights ($\hat{\beta}$), those weights giving cost functions minimized by empirical fin biases ($\alpha^* = \hat{\alpha}$) empirical fin biases, were estimated by linear interpolation. Confidence estimates on $\hat{\beta}$ were similarly interpolated from 95% confidence intervals of $\alpha^*$ evaluated at 95% confidence intervals of $\hat{\alpha}$.

