## [Decision Letter]

Thank you for submitting your article "A primal role for balance in the development of coordinated locomotion" for consideration by *eLife*. Your article has been reviewed by three peer reviewers, including Jennifer L Raymond ad the Reviewing Editor and Reviewer #1, and the evaluation has been overseen by Ronald Calabrese as the Senior Editor. The following individuals involved in review of your submission have agreed to reveal their identity: Kimberly McArthur (Reviewer #2); Hans Straka (Reviewer #3).

The reviewers have discussed the reviews with one another and the Reviewing Editor has drafted this decision to help you prepare a revised submission.

Summary:

This study employed behavioral analysis to explore the coordination of locomotion and posture in larval zebrafish. The results reveal a developmental change in the coordination between axial musculature and pectoral fins during swimming, with a greater usage of pectoral fins to generate lift in older animals, and increasingly more stable (horizontal) posture. A key finding is that vestibular signals from the utricular otoliths, which sense pitch, are critical to the coordinated engagement of pectoral fins to generate lift during upward swimming. In addition, the results indicate that the cerebellum functions to suppress pectoral fin recruitment, especially during "nose"-down swimming when pectoral recruitment would be antagonistic, but also during nose-up swimming. Computational modeling is used to explore explanations for the developmental plasticity in the coordination of body and fin movements.

This is a unique piece of work. The authors' description of the vertical motion of zebrafish in the water column is meticulous, providing a convincing platform for all further experiments. The results establish a tractable model for studying the neural substrates of the development of motor coordination, in which: 1) component behaviors are already functional, but their relative deployment changes with age; and 2) motor coordination impacts posture, but in a simpler way than typically occurs in terrestrial vertebrates.

Essential revisions:

1) It is not clear what the model adds. The proposal regarding a trade-off between effort and "balance" is intriguing, but the reviewers were not convinced that the modeling rigorously supports it, and wondered whether the paper might actually be stronger without the modeling. Though not immediately clear from the Results as written, the increased modeled effort during fin-biased swims seems to be driven by an increase in the number of maneuvers needed to reach a target. This seems to assume that the fish does not compensate for its motor strategy when executing an upward swim – that it will plan a given trajectory change in pitch, but always fall short of that change whenever fin bias is nonzero. Since the empirical behavioral data was recorded during spontaneous swims, there is no available data to support that assumption.

The authors should consider removing the model from the manuscript, and include the idea of a potential trade-off between effort and posture in the Discussion as compelling speculation, grounded in the rigorous behavioral data. At the very least, the description and the associated discussion of the model should be streamlined considerably.

2) The authors would like to conclude that "balance" plays a fundamental role in the development of fin-body coordination. The word "balance" is used throughout, and this word lacks some important precision (though it may enhance readability). If the authors intend "balance" to mean the sense of balance from the vestibular system, then the big conclusion (stated in the title) is consistent with the data: abolishing the otolith organs also abolishes normal fin-body coordination, consistent with a role for the vestibular system in development and/or execution of coordination. However, it seems that the authors intend "balance" to mean the drive to maintain a horizontal (upright) posture. This is a more provocative (and interesting) proposal, but it seems to be a step beyond what can be definitively concluded from the data. Indeed, the authors state it as a proposal in the Introduction (which is quite well-written), but employ it as a conclusion in the title and Discussion.

3) Currently the authors use a very technical language that makes it rather difficult to link the descriptions of e.g. slopes and sigmoid shapes with a particular behavioral parameter. Even though the "attack angle" was properly described in the beginning, it is still a very vague term that might be better explained throughout the manuscript by using additional icons in each figure. Also, the very detailed and often complicated descriptions impair grasping the major take-home message despite the summarizing sentences at the end of each part.

4) Though several possibilities are sprinkled throughout the text, the Discussion would benefit from a clear laying-out of the possible implications of the study (which are really interesting), regarding developmental changes in the animal's ability to generate optimal coordinated behaviors and/or changes in what the optimal coordinated behavior actually entails (more/less fin bias). If the definition of "optimal" doesn't change, then the animal's ability to execute it must be changing (possibly due to immature vestibular input, as suggested by the authors in the eighth paragraph of the Discussion; possibly due to immature coordination circuitry somewhere else; probably not due to changes in motor ability, from the study's results). If the ability to execute doesn't change, then the definition of "optimal" must be changing. The authors allude to the possibility that maintaining horizontal posture might be increasingly related to performance with age (Discussion, ninth paragraph), but don't discuss further. Articulating a clear theoretical framework in the Discussion would strengthen an already strong paper, by providing a blueprint for future experiments.

Title: As discussed above (see point #2) the conclusion stated in the title can be taken as more of a proposal by the authors, rather than a summary of what they've demonstrated. The title should be rewritten to for clarity, to disambiguate the dual use of the word "balance."

[Editors' note: further revisions were requested prior to acceptance, as described below.]

Thank you for submitting your article "A primal role for the sense of balance in the development of coordinated locomotion" for consideration by *eLife*. Your article has been evaluated by Ronald Calabrese as the Senior Editor and a Reviewing Editor.

The Reviewing editor has drafted this decision to help you prepare a revised submission.

Essential revisions:

The authors were responsive to the reviewers' comments, and the manuscript has been greatly improved. However, two points could be better addressed:

Previous point 2: The word "balance" is used throughout, and this word lacks some important precision (though it may enhance readability). If the authors intend "balance" to mean the sense of balance from the vestibular system, then the big conclusion (stated in the title) is consistent with the data: abolishing the otolith organs also abolishes normal fin-body coordination, consistent with a role for the vestibular system in development and/or execution of coordination. However, it seems that the authors intend "balance" to mean the drive to maintain a horizontal (upright) posture. This is a more provocative (and interesting) proposal, but it seems to be a step beyond what can be definitively concluded from the data. Indeed, the authors state it as a proposal in the Introduction (which is quite well-written), but employ it as a conclusion in the title and Discussion.

Response: "For clarity, we now utilize the term "sense of balance" in the Title and have emphasized the importance of sensation (as opposed to motor goals) for regulating coordination in the Abstract and Discussion (specifically Discussion first and fifth paragraphs)."

These revisions have not adequately addressed the lack of clarity resulting from use of "balance" in two different ways. I get that you are trying to distinguish "balance sense" from "balance performance", but this subtlety will undoubtedly be lost on readers who do not read as carefully. It is not clear why you are so reluctant to use the term "vestibular", which could help. Another term that would help in many places where you use the term balance "in the pitch axis." Second best would be to explicitly define and distinguish balance sense from balance performance somewhere very early in the manuscript. Some of the places where ambiguity persists, Abstract, Introduction, third and last paragraphs, subsection “Developing larvae regulate fin-body coordination”, last paragraph and in the subsection “Increasing fin bias improves balance but costs effort in silico”. In the aforementioned subsection, 'balance' means something different in the two successive sentences.

Previous point 4: Though several possibilities are sprinkled throughout the text, the Discussion would benefit from a clear laying-out of the possible implications of the study (which are really interesting), regarding developmental changes in the animal's ability to generate optimal coordinated behaviors and/or changes in what the optimal coordinated behavior actually entails (more/less fin bias). If the definition of "optimal" doesn't change, then the animal's ability to execute it must be changing (possibly due to immature vestibular input, as suggested by the authors in the eighth paragraph of the Discussion; possibly due to immature coordination circuitry somewhere else; probably not due to changes in motor ability, from the study's results). If the ability to execute doesn't change, then the definition of "optimal" must be changing. The authors allude to the possibility that maintaining horizontal posture might be increasingly related to performance with age (Discussion, ninth paragraph), but don't discuss further. Articulating a clear theoretical framework in the Discussion would strengthen an already strong paper, by providing a blueprint for future experiments.

Response: "We share the reviewers' interest in these questions, and appreciate the rich context treating the development of coordination as an optimization problem. The implications noted here were originally framed with respect to the model, which is the only part of the study that directly speaks to questions of optimality. To address the first point in the Revisions, we have moved the model to a separate Appendix. Therefore we expound on these ideas in a new paragraph added to the Appendix ("Conclusions about optimizing coordination"). "

A few lines about these ideas in the Discussion would be helpful. The model can stay in the Appendix, and be referenced.

---

## [Author Response]

Essential revisions:1) It is not clear what the model adds. The proposal regarding a trade-off between effort and "balance" is intriguing, but the reviewers were not convinced that the modeling rigorously supports it, and wondered whether the paper might actually be stronger without the modeling.

The rules governing the optimization of motor coordination in general, and of fin-body coordination specifically, are of particular interest. However, identifying these rules is challenging to assess empirically. We recognize that the simulations are largely speculative and include a number of assumptions in order to define and model the effort involved in swimming. Despite its speculative nature, the model constitutes a first foray into addressing these optimization rules and makes one critical, tangible contribution: it offers a framework to guide future studies on the neural mechanisms for experience-dependent regulation of fin-body coordination. To streamline the manuscript and focus on empirical data, we have moved the model and associated text to an Appendix.

Though not immediately clear from the Results as written, the increased modeled effort during fin-biased swims seems to be driven by an increase in the number of maneuvers needed to reach a target. This seems to assume that the fish does not compensate for its motor strategy when executing an upward swim – that it will plan a given trajectory change in pitch, but always fall short of that change whenever fin bias is nonzero. Since the empirical behavioral data was recorded during spontaneous swims, there is no available data to support that assumption.The authors should consider removing the model from the manuscript, and include the idea of a potential trade-off between effort and posture in the Discussion as compelling speculation, grounded in the rigorous behavioral data. At the very least, the description and the associated discussion of the model should be streamlined considerably.

This is absolutely correct. Because fin-dominant swimming cannot achieve climbs as steep as coordinated swims, more swims are needed to reach the same target. We now make this point clear in the simulation results presented in the Appendix.

2) The authors would like to conclude that "balance" plays a fundamental role in the development of fin-body coordination. The word "balance" is used throughout, and this word lacks some important precision (though it may enhance readability). If the authors intend "balance" to mean the sense of balance from the vestibular system, then the big conclusion (stated in the title) is consistent with the data: abolishing the otolith organs also abolishes normal fin-body coordination, consistent with a role for the vestibular system in development and/or execution of coordination. However, it seems that the authors intend "balance" to mean the drive to maintain a horizontal (upright) posture. This is a more provocative (and interesting) proposal, but it seems to be a step beyond what can be definitively concluded from the data. Indeed, the authors state it as a proposal in the Introduction (which is quite well-written), but employ it as a conclusion in the title and Discussion.

For clarity, we now utilize the term “sense of balance” in the Title and have emphasized the importance of sensation (as opposed to motor goals) for regulating coordination in the Abstract and Discussion (specifically Discussion, first and fifth paragraphs).

3) Currently the authors use a very technical language that makes it rather difficult to link the descriptions of e.g. slopes and sigmoid shapes with a particular behavioral parameter. Even though the "attack angle" was properly described in the beginning, it is still a very vague term that might be better explained throughout the manuscript by using additional icons in each figure. Also, the very detailed and often complicated descriptions impair grasping the major take-home message despite the summarizing sentences at the end of each part.

To aid reading, we have added iconography to Figures 1B, 1C, (new explanatory panel), 2E, and 2F. We have changed wording and added explanatory sentences in lay language throughout the manuscript. We also use the term “fin-body ratio” to describe the sigmoid maximal slope.

4) Though several possibilities are sprinkled throughout the text, the Discussion would benefit from a clear laying-out of the possible implications of the study (which are really interesting), regarding developmental changes in the animal's ability to generate optimal coordinated behaviors and/or changes in what the optimal coordinated behavior actually entails (more/less fin bias). If the definition of "optimal" doesn't change, then the animal's ability to execute it must be changing (possibly due to immature vestibular input, as suggested by the authors in the eighth paragraph of the Discussion; possibly due to immature coordination circuitry somewhere else; probably not due to changes in motor ability, from the study's results). If the ability to execute doesn't change, then the definition of "optimal" must be changing. The authors allude to the possibility that maintaining horizontal posture might be increasingly related to performance with age (Discussion, ninth paragraph), but don't discuss further. Articulating a clear theoretical framework in the Discussion would strengthen an already strong paper, by providing a blueprint for future experiments.

We share the reviewers’ interest in these questions, and appreciate the rich context treating the development of coordination as an optimization problem. The implications noted here were originally framed with respect to the model, which is the only part of the study that directly speaks to questions of optimality. To address the first point in the revisions, we have moved the model to a separate Appendix. Therefore we expound on these ideas in a new paragraph added to the Appendix (“Conclusions about optimizing coordination”).

Title: As discussed above (see point #2) the conclusion stated in the title can be taken as more of a proposal by the authors, rather than a summary of what they've demonstrated. The title should be rewritten to for clarity, to disambiguate the dual use of the word "balance."

Fixed, as detailed above.

[Editors' note: further revisions were requested prior to acceptance, as described below.]

Essential revisions:The authors were responsive to the reviewers' comments, and the manuscript has been greatly improved. However, two points could be better addressed:Previous point 2: The word "balance" is used throughout, and this word lacks some important precision (though it may enhance readability). If the authors intend "balance" to mean the sense of balance from the vestibular system, then the big conclusion (stated in the title) is consistent with the data: abolishing the otolith organs also abolishes normal fin-body coordination, consistent with a role for the vestibular system in development and/or execution of coordination. However, it seems that the authors intend "balance" to mean the drive to maintain a horizontal (upright) posture. This is a more provocative (and interesting) proposal, but it seems to be a step beyond what can be definitively concluded from the data. Indeed, the authors state it as a proposal in the Introduction (which is quite well-written), but employ it as a conclusion in the title and Discussion.Response: "For clarity, we now utilize the term "sense of balance" in the Title and have emphasized the importance of sensation (as opposed to motor goals) for regulating coordination in the Abstract and Discussion (specifically Discussion, first and fifth paragraphs)."These revisions have not adequately addressed the lack of clarity resulting from use of "balance" in two different ways. I get that you are trying to distinguish "balance sense" from "balance performance", but this subtlety will undoubtedly be lost on readers who do not read as carefully. It is not clear why you are so reluctant to use the term "vestibular", which could help. Another term that would help in many places where you use the term balance "in the pitch axis." Second best would be to explicitly define and distinguish balance sense from balance performance somewhere very early in the manuscript. Some of the places where ambiguity persists, Abstract, Introduction, third and last paragraphs, subsection “Developing larvae regulate fin-body coordination”, last paragraph and in the subsection “Increasing fin bias improves balance but costs effort in silico”. In the aforementioned subsection, 'balance' means something different in the two successive sentences.

We have made the requested clarifications and augmented the title similarly.

Previous point 4: Though several possibilities are sprinkled throughout the text, the Discussion would benefit from a clear laying-out of the possible implications of the study (which are really interesting), regarding developmental changes in the animal's ability to generate optimal coordinated behaviors and/or changes in what the optimal coordinated behavior actually entails (more/less fin bias). If the definition of "optimal" doesn't change, then the animal's ability to execute it must be changing (possibly due to immature vestibular input, as suggested by the authors in the eighth paragraph of the Discussion; possibly due to immature coordination circuitry somewhere else; probably not due to changes in motor ability, from the study's results). If the ability to execute doesn't change, then the definition of "optimal" must be changing. The authors allude to the possibility that maintaining horizontal posture might be increasingly related to performance with age (Discussion, ninth paragraph), but don't discuss further. Articulating a clear theoretical framework in the Discussion would strengthen an already strong paper, by providing a blueprint for future experiments.Response: "We share the reviewers' interest in these questions, and appreciate the rich context treating the development of coordination as an optimization problem. The implications noted here were originally framed with respect to the model, which is the only part of the study that directly speaks to questions of optimality. To address the first point in the Revisions, we have moved the model to a separate Appendix. Therefore we expound on these ideas in a new paragraph added to the Appendix ("Conclusions about optimizing coordination"). "A few lines about these ideas in the Discussion would be helpful. The model can stay in the Appendix, and be referenced.

A new paragraph (“Considering the development of coordination and optimization process…”) outlines these ideas in the main text Discussion.